# State-selective small molecule degraders that preferentially remove aggregates and oligomers

Jakub Luptak[1,4,5], Dean Clift[1,5], Aamir Mukadam[2], Jonathan Benn[2], Tyler Rhinesmith [1], Stephen H. McLaughlin [1], Amy C. Dodds[3], Jerson E. Lapetaje[3], Matylda Sczaniecka-Clift[1], David J. France [3], William A. McEwan [2] & Leo C. James [1] ✉

TRIM21 is a unique E3 ligase that uses a clustering-based activation mechanism to degrade complex multimeric substrates. This activity underpins the targeted protein degradation technology Trim-Away and genetically encoded degraders that selectively target aggregated tau protein and prevent tauopathy. Here we describe small molecules that mimic TRIM21's natural epitope and function as either effective inhibitors or potent and selective degraders called TRIMTACs. TRIMTACs mediate degradation as rapidly as PROTACs but can also selectively degrade specific protein pools depending on assembly state. We demonstrate the utility of this state-specific degradation by selectively removing the pro-inflammatory signalling protein Myd88 when assembled into the Myddosome and the cell-death protein RIPK3 when polymerised into the Necrosome. We further show that TRIMTACs can inhibit seeded tau aggregation under conditions where a PROTAC is ineffective. These results highlight that TRIM21's clustering-based activation can be exploited by small molecule degraders to carry out state-selective degradation of therapeutic targets.

E3 ubiquitin ligases are responsible for transferring ubiquitin to target proteins, resulting in diverse cellular outcomes[1]. One key role of ubiquitination is to label substrates for degradation in the proteasome or by autophagy. The most well-studied family of E3s is the cullin-RING ligases (CRLs), which are modular complexes of independent scaffold, adaptor and ligase proteins that position specific substrates for ubiquitin transfer[2]. TRIM ligases are the largest family of E3 enzymes in mammals, but unlike CRLs, do not employ a modular system[3]. Instead, they are multidomain proteins that combine substrate binding and E3 ligase activities into a single polypeptide sequence. The unique architecture of TRIM proteins yields a single degradation-inducing enzyme that relies on direct substrate engagement for activation, and thus a

distinct mechanism compared to CRLs[4]. While conserved in their architecture, TRIM proteins function in highly diverse cellular pathways[5]. This includes antiviral responses[6], immunity[7,8], inflammatory disease[5], cancer, and tissue-specific functions[9,10] in the brain[11] and neurobiology. However, as TRIMs have not been as intensively studied as CRLs, there are many members of the family whose function is still unknown or is at least unclear. Recent work has also highlighted that some members are catalytically dead[12] and may have functions analogous to inactive regulatory pseudoproteases[13].

TRIM21 is an archetypal TRIM protein, first discovered over 15 years ago as the E3 ligase and cytosolic IgG receptor responsible for intracellular antibody immunity[14,15]. TRIM21 intercepts antibody-

[1]MRC Laboratory of Molecular Biology, Francis Crick Avenue, Cambridge, UK. [2]UK Dementia Research Institute at the University of Cambridge, Department of Clinical Neurosciences, Hills Road, Cambridge, UK. [3]School of Chemistry, Joseph Black Building, University of Glasgow, Glasgow, UK. [4]Present address: Protein Sciences, Structure, and Biophysics, Discovery Sciences, R&D, AstraZeneca, Cambridge, UK. [5]These authors contributed equally: Jakub Luptak, Dean Clift. ✉e-mail: lcj@mrc-lmb.cam.ac.uk

coated viruses that enter the cell and targets them for rapid degradation[15]. The need to degrade large complex structures quickly and efficiently has driven the evolution of a ubiquitination mechanism with very specific properties. TRIM21 is broadly expressed[16] in many cell types[17] and tissues[18] but in an inactive state[19]. This is due to autoinhibition of the RING domain by the B Box domain, which blocks the E2 binding site[19], and by the protein's homodimeric domain architecture that prevents constitutive RING dimerisation by separating each RING domain at either end of an extended anti-parallel coiled-coil domain[20]. TRIM21 is activated by a substrate-induced clustering mechanism where the binding of multiple TRIM21 molecules allows inter-molecular RING dimerisation and activation of ubiquitination[20–22]. Importantly, this imbues TRIM21 with intrinsic mechanistic specificity for oligomers, because there will be no clustering on monomeric targets and hence no activation. Moreover, this system ensures that ubiquitination activity increases with substrate multimericity: the larger and more polymeric the substrate, the larger the number of TRIM21 molecules that are recruited and activated[20]. This ability for TRIM21 to transform highly ordered and complex substrates into signal amplification platforms underpins its antiviral function by allowing it to degrade highly challenging targets such as viral capsids and multivalent ribonucleoprotein complexes from diverse pathogens, including adenovirus[15], rotavirus[23], picornavirus[24], and bunyaviruses such as Crimean-Congo haemorrhagic fever virus and lymphocytic choriomeningitis virus[25–27].

TRIM21s intrinsic mechanistic specificity has also been exploited to carry out targeted protein degradation. In the technology 'Trim-Away', off-the-shelf antibodies are electroporated into cells and used to recruit endogenous TRIM21 to desired protein targets[28]. The resulting degradation is specific to oligomers and rapid, occurring within a few hours. For instance, monomeric GFP is not degraded in the presence of a monoclonal anti-GFP antibody[20], whereas the highly oligomeric protein NUP98 is efficiently removed by anti-NUP98 antibodies[28]. A crucial advantage of TRIM21s clustering-based activation is that it can be used to degrade specific protein states whilst leaving other forms of the same or similar protein untouched. In Huntington's disease, two forms of huntingtin protein are expressed—one wild-type copy with a short polyQ sequence, and an expanded polyQ repeat-bearing version that exerts toxic gain of function. When recruited by anti-polyQ antibodies, TRIM21 degrades the expanded protein but leaves the wildtype protein intact because only the former recruits sufficient TRIM21 to trigger clustered activation[20,28]. Furthermore, the longer the polyQ repeat, the more efficient the degradation[20]. This same mechanism-driven specificity has been exploited to facilitate the degradation of tau aggregates whilst leaving functional tau protein intact[29,30]. Indeed, TRIM21-mediated degradation has been shown to be the mechanism behind effective immunotherapy in a mouse model of tauopathy[30].

Degrader molecules made by fusing the TRIM21 RING domain to a targeting domain, such as a nanobody, preserve this clustering mechanism and have been used to selectively degrade only multivalent substrates[20,31,32]. This approach has been exploited in the design of degrader molecules against tau aggregates, in which the TRIM21 RING domain was fused to either an anti-tau nanobody (RING-nanobody[33]) or tau protein itself (RING-bait[34]). In both cases, the TRIM21-based degrader is designed to activate only when the RING-nanobody or RING-bait is recruited into aggregates. When blood-brain barrier crossing adeno-associated viruses (AAVs) are used to deliver the gene for these degraders directly into the brain in a P301S tauopathy mouse model, levels of tau aggregates are reduced without any change in overall tau protein levels[33,34]. Most importantly, treated mice also show improved behavioural signs, including preservation of motor function[34]. The above studies highlight that TRIM21 possesses mechanistic properties with important implications for targeted protein degradation applications, and which distinguish them from CRLs.

A popular modality for targeted protein degradation are small molecule degraders, or 'PROTACs', (proteolysis targeting chimeras)[35,36] but as these generally use CRLs they are not capable of the selective degradation achievable when recruiting TRIM21.

Here, we present specific TRIM21 ligands that can be made into heterobifunctional degraders called TRIMTACs to recruit endogenous TRIM21 and selectively degrade proteins in specific functional states. We show that TRIMTACs are capable of selectively degrading signalling proteins Myd88 and RIPK3 only when they are assembled into active complexes, driving inflammation and necroptosis, respectively. Furthermore, we compare TRIMTAC and PROTAC degraders and show that only the former is capable of efficiently inhibiting seeded tau aggregation.

## Results

### Small molecules can mimic antibody binding to TRIM21 and inhibit its antiviral and degradative activity

We used a DNA-encoded library (DEL) screening approach to obtain small-molecule ligands that bind to the PRYSPRY domain of TRIM21. TRIM21 PRYSPRY was directly derivatized to carboxyl MagnaBind beads, with pull-down experiments against IgG and competition with protein AG confirming that the binding site remains exposed (Supplementary Fig. 1a). T21 PRYSPRY-beads were screened against the WuXi DEL Open library, consisting of 4 billion compounds. Three hits (MRC36, MRC37 and MRC38) were synthesised off-DNA and tested for binding to TRIM21 PRYSPRY by thermal stability and Tryptophan quenching, with fluorescence polarisation used to determine accurate affinity measurements (Supplementary Fig. 1b–e). MRC37 and MRC38 stabilised TRIM21 PRYSPRY the most, and this correlated with their greater affinity (0.4 μM and 1.8 μM, respectively). All three compounds competed with IgG Fc, suggesting an overlapping binding site (Supplementary Fig. 1e). We determined crystal structures of each compound in complex with TRIM21 PRYSPRY and compared them to the published complex with IgG Fc (Fig. 1a, Supplementary Table 1 and Supplementary Fig. 1f). All three compounds occupy the IgG Fc binding pocket and overlap with the natural 'HNHY' binding epitope (Fig. 1b). In particular, MRC37 and MRC38 closely mimic the bidendate engagement of Fc-H433 and Fc-N434 deep within the pocket, with the L-4-Pal of MRC37 occupying the position of Fc-H433 and the cyclohexyl-proline filling the position of Fc-N434. Similar to IgG Fc, MRC37 and MRC38 are reliant on key PRYSPRY residue W381, whose mutation to alanine decreases binding 100-fold (Supplementary Fig. 1g). Surprisingly, however, neither compound was significantly affected by mutations W383A and D355A, which abolish Fc binding. This result is also significant given that D355 limits binding to a recently reported acepromazine ligand for TRIM21, with mutation D355A necessary to improve binding and efficacy of the compound[37]. These compounds are therefore more potent ligands for wild-type TRIM21 and are independent of the alanine mutation at this position. While W381A is critical for MRC37 and MRC38 binding, there are other residues in proximity to the ligands that may contribute to binding, including Y328, F450, M330, S447, L371, L370, F369 and H368.

Previously reported TRIM21 ligands are cytotoxic because they act as a molecular glue between TRIM21 and Nup98, causing degradation of the latter and cell death[37–41]. We therefore tested MRC37 for cytotoxicity, using TRIM21 glue compound PRLX-93936 as a positive control. There was no evidence of cytotoxicity during 48 h treatment with 5 μM MRC37, as measured by cell confluence (Fig. 1c). Increasing the concentration to 20 μM also had no affect on either cell confluence or cell viability (Fig. 1d, e). In contrast, cell confluence decreased rapidly after -12 h treatment with 5 μM PRLX-93936, and both confluence and cell viability were <50% after 48 h incubation with concentrations > 1 μM (Fig. 1c–e). Importantly, PRLX-93936 cytotoxicity was abrogated in TRIM21 knockouts, consistent with its reported

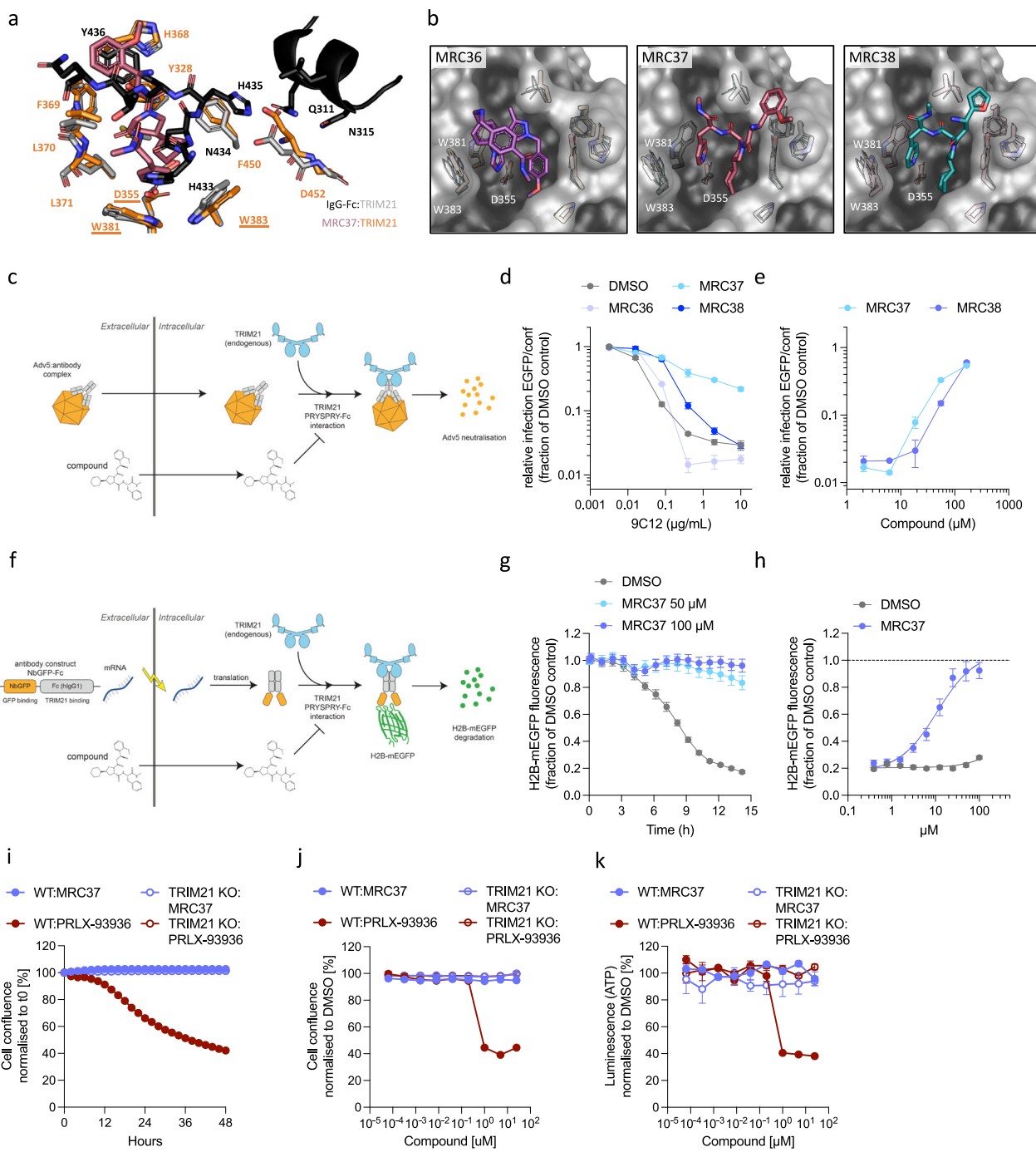

Nup98 glue mechanism. These results indicate that MRC37 does not have the same on-target cytotoxicity as other reported TRIM21 ligands.

Next, we tested whether these compounds could inhibit natural antiviral function and the ability to degrade proteins. TRIM21 mediates its antiviral effects by mediating the degradation of cytosolic viruses coated in antibodies. This requires binding of the PRYSPRY domain to the Fc of the antibodies attached to the surface of the virus. To test inhibition of antiviral function, we treated HEK293T cells with 50 µM compounds or DMSO, then infected with human adenovirus serotype 5-EGFP (Adv5) in the presence of antibody 9C12, which is reported to neutralise Adv5 through TRIM21 in the cytosol (Fig. 1f). If the compounds are cell permeable and bind to PRYSPRY in endogenous TRIM21, this should lead to loss of neutralisation. MRC37 substantially reduced neutralisation, with MRC38 having an intermediate effect and MRC36 showing negligible activity (Fig. 1g). When we held the 9C12

concentration constant, we observed that inhibition of neutralisation by MRC37 and MRC38 increased with dose, highlighting that inhibition is likely specific and not the result of toxicity (Fig. 1h). To test the ability of our compounds to block TRIM21-mediated degradation directly we performed a modified Trim-Away experiment in which we electroporated RPE-1 cells expressing H2B-mEGFP with mRNA encoding a nanobody-Fc fusion against mEGFP (NbGFP-Fc). Expression of the NbGFP-Fc results in a complex with H2B-mEGFP and TRIM21, causing H2B-mEGFP degradation (Fig. 1i). Degradation following electroporation is rapid and efficient, with H2B-mEGFP levels beginning to decrease after 3 h and a loss of ~80% of the fluorescent signal by 12 h (Fig. 1j). In contrast, in cells treated with MRC37, there was very little loss of H2B-mEGFP fluorescence even after 15 h (Fig. 1k). Performing a titration of MRC37 revealed that as in the antiviral assay, inhibition of TRIM21 degradation function was dose-dependent. Taken together,

**Fig. 1 | Small molecules can mimic antibody binding to TRIM21 PRYSPRY (T21PY) and inhibit its antiviral and degradative activity. a** Comparison of the complex between T21PY:IgG Fc and T21PY:MRC37. All residues are shown as stick representation with the IgG Fc helix (residues 308-317) shown as a cartoon representation. **b** Crystal structures of T21PY in complex with MRC36-38. T21PY is shown as a surface representation in grey, with selected residues shown as stick representations. Residue positions from the other complexes are shown as line representations. **c** DMSO, 5 μM MRC37 or 5 μM PRLX-93936 was added to WT or TRIM21 KO RPE-1 cells and cell confluence was monitored over 48 h. **d** As (c), except a range of compound concentrations and confluence measured at 48 h. **e** As (d), except cell viability measured by Luminescent ATP Detection. **c**–**e** Graphs show data from $2 \times 10^4$ cells and are normalised to the DMSO control. Data is expressed as mean and s.e.m. from n = 4 technical replicates. Representative examples from n = 2 independent experiments. **f** Schematic of an adenovirus neutralisation experiment. TRIM21 uses anti-adenovirus antibodies (9C12) to prevent infection. Membrane-permeable compounds that bind endogenous TRIM21 would be expected to restore infection. **g** Adenovirus neutralisation experiment measuring infection in the presence of DMSO or 50 μM compounds. **h** Adenovirus neutralisation titrating MRC37 and MRC38. (d&e) Infection is calculated as the fraction of total cells that are EGFP positive (from $1 \times 10^4$ cells). Graphs show mean and s.e.m. from n = 3 technical replicates. Representative examples from n = 3 independent experiments. **i** Schematic of TRIM21-mediated degradation assay. Expression of an NbGFP-Fc antibody recruits TRIM21 to H2B-mEGFP, causing H2B-mEGFP proteasomal degradation. MRC37 competes with NbGFP-Fc for binding to TRIM21 PRYSPRY, thus inhibiting TRIM21 recruitment and H2B-mEGFP degradation. **j** Time course of H2B-mEGFP fluorescence after electroporation of NbGF-Fc mRNA and concomitant addition of compound at the indicated concentrations. **k** Dose-response curve of rescue of H2B-mEGFP degradation at 12 h post-electroporation of NbGFP-Fc mRNA and concomitant addition of MRC37 or matched DMSO concentrations. (j&k) Graphs show the integrated density of GFP fluorescence (in relative fluorescence units, RFU) normalised to total cell area (phase) from $1 \times 10^4$ cells and expressed as a fraction of the DMSO control. Data is expressed as mean and s.e.m. from n = 4 technical replicates. Representative examples from n = 2 independent experiments. Source data are provided as a Source Data file.

these results show that compounds that occupy the Fc-binding site on TRIM21 PRYSPRY and mimic the 'HNHY' epitope are capable of inhibiting TRIM21s antiviral and degradative functions in the cell.

## TRIMTACs drive ternary complex formation in vitro and in cells

We next sought to convert our small molecule inhibitors into bispecific ligands capable of functioning like PROTACs, or in this case, 'TRIMTACs' – Tripartite motif protein-engaging targeting chimeras. We chose to focus on MRC37 as this had the highest binding affinity for TRIM21 PRYSPRY and because we identified three possible exit vectors using our crystal structure where additional moieties could be easily added (Fig. 2a). These included V1, where the methoxy group of the benzene points into solvent, V2 where an extra acid can be readily coupled during synthesis, and the former DNA attachment point V3. Modified ligands were synthesised with either a short PEG linker (V1), a histidine amino acid (V2) or a PEG linker with chloroalkane (V3) and docked onto the existing structure (Fig. 2b, Supplementary Fig. 1f). Thermal stability measurements confirmed that binding is preserved in each case (Fig. 2c). MRC71 (MRC71), containing the PEG linker and chloroalkane was selected for further experiments as the latter moiety allows covalent binding to HaloTag protein. We used analytical ultracentrifugation to test whether the 11-atom PEG linker in MRC71 is sufficient to allow a ternary complex to form between TRIM21 PRYSPRY and HaloTag in vitro. In the presence of MRC71, we observed loss of peaks corresponding to individual TRIM21 PRYSPRY and HaloTag (22 kDa and 34.4 kDa mass respectively) and gain of a new peak with a mass of 57.3 kDa, close to that predicted for the ternary complex (Fig. 2d). We extended this approach and titrated MRC71, observing a hook effect[42] in which the average peak mass approached that expected for the ternary complex at equimolar concentrations before decreasing under conditions of compound excess (Fig. 2e). With in vitro data supportive of ternary complex formation, we next tested whether MRC71 could recruit full-length TRIM21 to a HaloTag-containing target in cells. To this end, we exploited an RPE1 cell line transiently over-expressing mCherry-TRIM21 and H2B-mEGFP-Halo and used live cell imaging to detect co-localisation following the addition of either monovalent MRC37 or bispecific MRC71. Consistent with previous data showing that mCherry-TRIM21 is too large to enter the nucleus, neither compound initially altered protein localisation (Fig. 2f, 0 h timepoint). However, upon transient nuclear envelope breakdown during cell division, MRC71 induced an accumulation of mCherry-TRIM21 inside the nucleus (Figs. 2f, 1h). Moreover, mCherry-TRIM21 remained enriched within the nucleus upon completion of cell division, where it co-localised with H2B-mEGFP-Halo (Fig. 2f, h). In contrast, there was no co-localisation in MRC37-treated cells either before, during or after cell division (Fig. 2f). Having established a

proxy for ternary complex formation inside cells, we titrated each compound and observed dose-dependent co-localisation of mCherry-TRIM21 and H2B-mEGFP-Halo with MRC71 (Fig. 2g). There was also a hook effect that closely recapitulated in vitro data. Repeating the titration experiments in cells expressing different TRIM21 domain deletions, we confirmed that, as expected, the ability of MRC71 to drive co-localisation was independent of the RING and B-Box domains but dependent on the PRYSPRY (Fig. 2h, i). Linker length is a key parameter in bispecific small molecule design. To provide a structural basis for linker length optimisation, MRC71 was crystallised with HaloTag and the resulting structure used to model a ternary complex with TRIM21 PRYSPRY (Fig. 2j, Supplementary Table 1). On the basis of this model, we synthesised chloroalkane variants with PEG linkers between 8 and 18 atoms in length (Supplementary Fig. 2). These variants were titrated onto RPE1 cells stably overexpressing mCherry-TRIM21/H2B-mEGFP-Halo to increase the potential co-localisation signal and their activity compared. MRC109 and MRC111, with longer linker lengths of 14 and 18 atoms, respectively, behaved similarly to MRC71, albeit with a slightly delayed hook effect (Fig. 2k). These results show that a bispecific molecule with a ligand against TRIM21 and a ligand against HaloTag - or 'HaloTRIMTAC' - is capable of forming a ternary complex between ligase and substrate.

## TRIMTACs degrade oligomeric but not monomeric targets

Next, we tested whether our HaloTRIMTACs could recapitulate the oligomer-specific degradation of the same cellular targets as previously reported for antibodies in Trim-Away experiments. To allow a direct comparison, we used the same targets as in published Trim-Away assays, namely monomeric mEGFP, oligomeric CAV1-mEGFP and oligomeric Cavin-1 mEGFP, with the addition of a HaloTag to each construct (Fig. 3a). Imaging the resulting cell lines gave expression profiles consistent with diffuse cytosolic localisation for monomeric mEGFP-Halo and plasma membrane localisation for CAV1- and Cavin1-mEGFP-Halo, both of which form oligomers on caveoli membrane structures (Fig. 3b). We titrated our halo-TRIMTACs with different linker lengths onto these cells and measured target protein levels by GFP fluorescence after 48 h. To provide a direct comparison with a PROTAC equivalent small-molecule degrader, we tested in parallel a VHL-recruiting compound with a chloroalkane referred to here as 'haloPROTAC1'[43]. As expected, haloPROTAC1 efficiently degraded all three substrates−monomeric mEGFP-Halo and oligomeric CAV1-mEGFP-Halo and Cavin1-mEGFP-Halo (Fig. 3c). In contrast, the haloTRIMTACs only degraded the oligomeric substrates. At high compound concentrations, particularly of MRC71, there was an increase in fluorescent monomeric mEGFP-Halo that is likely the result of binding-induced protein

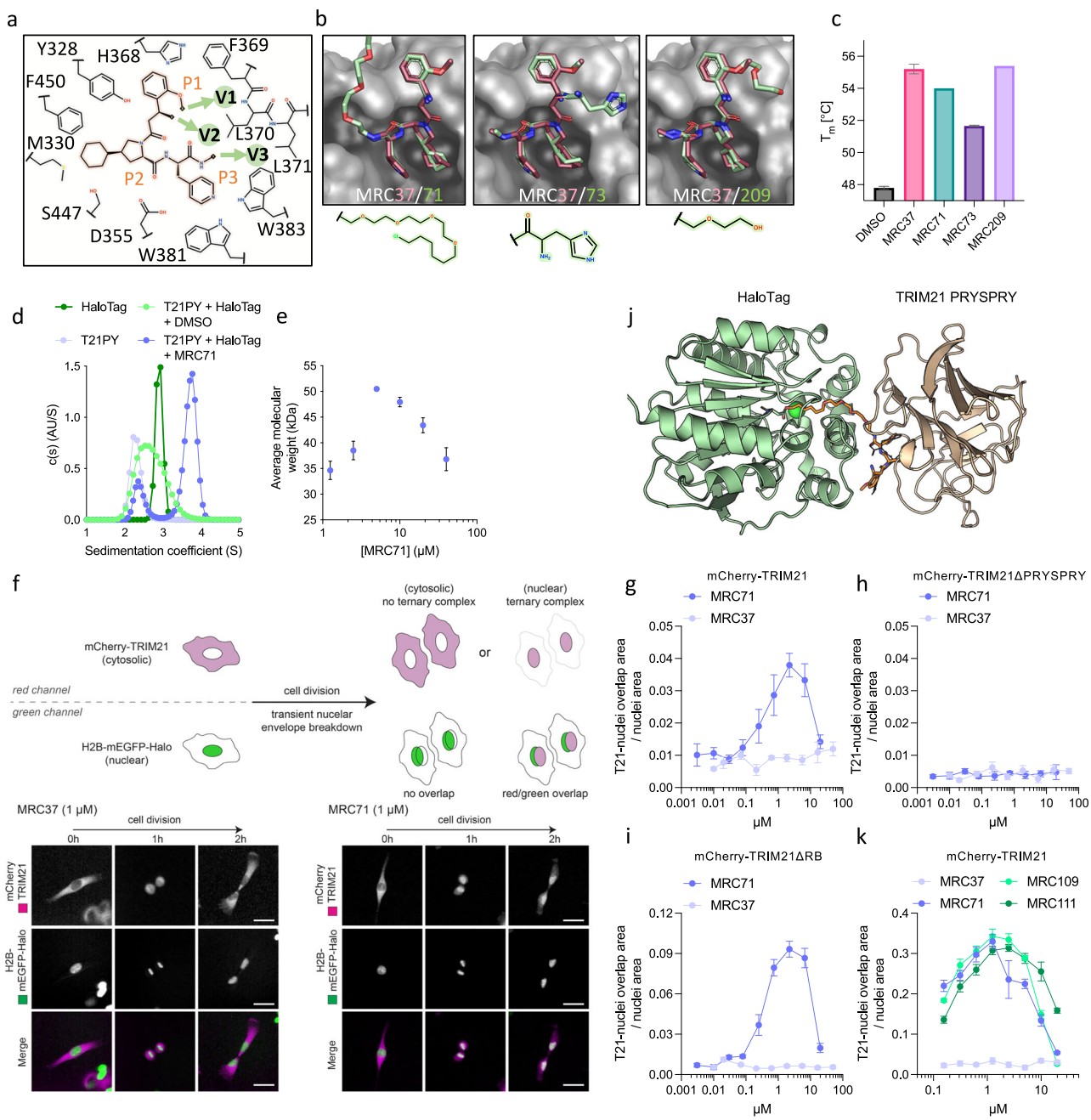

stabilisation. Both haloPROTAC1 and the haloTRIMTACs exhibited a hook effect, with maximum degradation observed at compound concentrations of ~ 1 μM (Fig. 3c). This pattern of degradation, including onset of hook effect, correlates closely with the ability of haloTRIMTACs to induce TRIM21:substrate co-localisation (Figs. 2k, 3c). Next, we compared the kinetics of degradation of haloPROTAC1 vs haloTRIMTACs. In each case, degradation proceeded until about 8 h when degradation plateaued (Fig. 3d). This suggests that the mechanism of degradation induced by TRIM21- or VHL-based degraders is similarly rapid. In both titration and time-course experiments, no degradation was observed with monovalent TRIM21 ligand MRC37. Degradation of both Cavin1-mEGFP-Halo and CAV1-mEGFP-Halo was abolished in TRIM21 KO cells, demonstrating that haloTRIMTACs induce degradation in a TRIM21-dependent manner (Fig. 3e, f). The oligomer-specific degradation achieved here by a small molecule degrader closely recapitulates that previously observed by antibodies on similar cellular targets[20,44].

## A haloTRIMTAC is more efficient than a VHL-based haloPROTAC at degrading an oligomeric substrate

Although they displayed similar degradation kinetics, our halo-TRIMTACs were apparently less efficient than a VHL-based haloPRO-TAC equivalent, with more total substrate remaining at the end of the experiment. To investigate this further, we examined the distribution of Cavin1-mEGFP-Halo inside cells following degradation. In untreated cells, Cavin1-mEGFP-Halo was observed diffusely in the cytosol and concentrated at the plasma membrane, consistent with a pool of unassembled protein and a pool of oligomeric protein at caveoli membranes. Following haloPROTAC1 treatment, levels of both cytoplasmic and membrane-associated Cavin1-mEGFP-Halo were reduced (Fig. 4a). In contrast, addition of haloTRIMTACs removed the membrane-associated Cavin1-mEGFP-Halo but left the diffuse cytosolic pool untouched (Fig. 4a). These data suggests that the apparent reduced efficiency of degradation by haloTRIMTACs vs haloPROTAC1 is actually because our haloTRIMTACs preserve the pool of soluble

**Fig. 2 | TRIMTACs drive ternary complex formation in vitro and in cells. a** 2D representation of the binding site with MRC37. Residues forming the pocket are shown as a skeletal representation. MRC37 is also shown as a skeletal representation but is highlighted in orange. P1–P3 indicate building blocks that make up MRC37 for potential substitution sites. Green arrows indicate potential exit vectors. **b** Models of indicated compounds superposed on MRC37:T21 PRYSPRY structure. **c** Thermal stabilisation of T21 PRYSPRY by the indicated compounds. Plotted melting temperature ($T_m$) is an average (± s.e.m.) of two independent replicates ($n = 2$). **d** AUC velocity experiment; The c(s) distributions show that both T21 PRYSPRY (solid black line) and HaloTag (dashed black line) alone are monomeric sedimenting at 2.3 S (**T**, $S_{w,20} = 2.5$ S) and 2.9 S (**H**, $S_{w,20} = 3.2$ S) respectively with calculated masses of 22.0 and 34.4 kDa with frictional ratios of 1.143 and 1.142 respectively. The mixture alone (orange dashed line) results in a single broad cumulative Guassian distribution representative of the two individual unbound species (**T−H**). Addition of ligand at 10 μM (solid orange line) reduces the concentration of T21 PRYSPRY and a large reduction in HaloTag with the appearance of a species sedimenting at 3.7 S (**T-71-H**, $S_{w,20} = 4.1$ S). This species has the expected mass of a ternary complex of 57.3 kDa, assuming a frictional ratio of 1.286 (characteristic of an extended non-spherical complex). **e** AUC equilibration calculated average mass ± s.d. for the T21 PRYSPRY ternary complex with HaloTag at a range of MRC37 concentrations. In the absence of the compound, the average mass of 30,316 Da is approximately the average of the two proteins at equimolar concentrations. As the concentration of the compound increases, the average mass reaches 50,482 ± 82 Da, close to the expected value for a 1:1 complex. Increasing concentration beyond this point decreases the average mass due to the binding of the compound to individual components in competition with complex formation. **f** Schematic and representative images of the cellular ternary complex assay. Cells co-express mCherry-TRIM21 and H2B-mEGFP-Halo. Presence of the large mCherry tag on TRIM21 prevents import into the nucleus. Upon cell division transient nuclear envelope breakdown allows access of mCherry-TRIM21 to chromatin. Nuclear localisation of mCherry-TRIM21 following cell division is indicative of ternary complex formation between mCherry-TRIM21, TRIMTAC and H2B-mEGFP-Halo. Scale bar = 30 μm. **g–i** HaloTRIMTAC (MRC71) exhibits a bell-shaped concentration-dependence (Hook effect) for ternary complex formation, which is dependent on TRIM21 PRYSPRY. **j** Comparison of ternary complex formation between HaloTRIMTACs with different linker lengths. **g–k** Graphs show the fraction of mCherry nuclear fluorescence that is also mEGFP positive from $1 \times 10^4$ cells. Data is expressed as mean and s.e.m. from $n = 4$ technical replicates. Representative examples from $n = 2$ independent experiments. **k** Model of the ternary complex between HaloTag (pale green) and T21 PRYSPRY (wheat) shown as cartoon representations, with a stick model of MRC71 covalently bonded to HaloTag. Generated from the structures of HaloTag:MRC71 and T21 PRYSPRY:MRC37. Source data are provided as a Source Data file.

Cavin1-mEGFP-Halo. Next, we performed a time-course experiment and compared haloPROTAC1 with haloTRIMTAC MRC71. The halo-TRIMTAC removed plasma-membrane-associated Cavin1-mEGFP-Halo rapidly and efficiently (Fig. 4b, Supplementary Fig. 3), but even after 24 h the diffuse cytoplasmic pool was unaffected. In contrast, the haloPROTAC1 removed the diffuse cytoplasmic pool, but significant membrane-associated Cavin1-mEGFP-Halo remained (Fig. 4b). This difference is also apparent when quantifying protein levels over time within a single cell. The haloPROTAC is poorly active against membrane-associated protein (Fig. 4c). In contrast, haloTRIMTAC1 efficiently degraded membrane-associated protein but left the diffuse cytoplasmic protein untouched (Fig. 4d). Using this approach, we also confirmed that loss of membrane-associated protein is via proteasomal degradation by using the inhibitor MG132. Addition of MG132 prevented substrate degradation by haloTRIMTAC MRC71 (Fig. 4e). In anti-GFP immunoblots, we observed that in addition to full-length Cavin1-mEGFP-Halo there is a smaller species likely corresponding to a cleavage product lacking Cavin1. The compound haloPROTAC1 efficiently degraded the smaller species but not the larger full-length version (Fig. 4f). In contrast, the haloTRIMTAC compound efficiently degraded the full-length protein with no impact on the smaller species (Fig. 4f). Taken together, the data show that a VHL-based PROTAC and a TRIMTAC can be functionally differentiated, with the latter displaying activity only for oligomeric substrates. Moreover, the tested haloTRIMTACs were more efficient at degrading oligomers than a haloPROTAC equivalent. This pattern of degradation specificity is consistent with the requirement for TRIM21 to undergo clustering before activation and the signal amplification that TRIM21 benefits from when clustering around oligomeric targets.

### TRIMTACs rapidly and selectively degrade Myddosome and Necrosome complexes

An advantage of TRIM21s unusual activation mechanism is that it allows the selective degradation of oligomeric but not monomeric states of the same substrate. One application where such specificity might be usefully deployed is the degradation of signalling complexes, which typically comprise large oligomeric platforms, whilst leaving unassembled components untouched. This is not possible using standard PROTAC molecules. To investigate this further, we tested whether we could use a TRIMTAC degrader to achieve selective degradation of two complexes - the Myddosome, formed by oligomeric Myd88, and the Necrosome, formed by oligomeric RIPK3. For these experiments, we synthesised a non-covalent version of our haloTRIMTAC in which the chloroalkane was replaced with a dTAG binding ligand (MRC414; 'dTAG-TRIMTAC'). We tested this dTAG-TRIMTAC in cells overexpressing either mEGFP-FKBP(F36V) or Cavin1-mEGFP-FKBP(F36V) (Supplementary Fig. 4a, b). Selectivity for the oligomeric substrate was preserved with rapid degradation kinetics (Supplementary Fig. 4c) and was TRIM21 dependent (Supplementary Fig. 4d). For Myddosome experiments, we used a previously published assay in which Myd88 is fused to GyrB, a domain from *E. coli* DNA gyrase that dimerises upon addition of the bivalent antibiotic coumermycin, allowing direct stimulation of Myddosome assembly. To this construct we also added a fluorescent reporter, mEGFP, and an FKBP(F36V) domain (Fig. 5a). Myd88-GyrB-mEGFP-FKBP(F36V) had a diffuse cytoplasmic distribution in stably expressing cells until the addition of coumermycin, which stimulated the formation of large clusters (Fig. 5b). In coumermycin-stimulated cells treated with the monovalent MRC37 these clusters persisted (Fig. 5b). In cells treated with MRC414, the dTAG-TRIMTAC, these clusters rapidly disappeared, until only non-assembled diffuse protein remained (Fig. 5b). This pattern was reflected in the quantification of total cell fluorescence, in which addition of MRC37 had little effect whilst MRC414 rapidly decreased part, but not all, the fluorescent signal (Fig. 5c). Importantly, there was no degradation of Myd88-GyrB-mEGFP-FKBP(F36V) by MRC414 in cells where Myddosome formation had not be stimulated by coumermycin (Fig. 5b; control). For the second signalling platform target, the necrosome, we made a similar construct, but where Myd88-GyrB was removed and replaced with RIPK3 at the C-terminus. In this case, a combination of TNF, Smac mimetic, and Z-VAD (T/S/Z) was used to simulate RIPK3 oligomerisation into the necrosome (Fig. 5d). In untreated control cells, there was no change in FKBP(F36V)-mEGFP-RIPK3 levels upon addition of MRC414. In contrast, there was a dose-dependent removal of FKBP(F36V)-mEGFP-RIPK3 in T/S/Z-stimulated cells (Fig. 5e, f). Moreover, the addition of MRC414 inhibited cell death, albeit not to the same extent as the kinase inhibitor GSK-872 (Fig. 5g, h). However, in contrast to cells co-treated with T/S/Z and MRC414, co-treatment of T/S/Z and GSK-872 did not prevent RIPK3 clusters from forming and cells were full of these large structures within 5 h (Supplementary Fig. 5). Taken together, data from both Myddosome and Necrosome assays illustrate that TRIMTACs can be used to selectively degrade signalling components only when they have assembled into complexes, with no degradation of the unassembled components in quiescent cells.

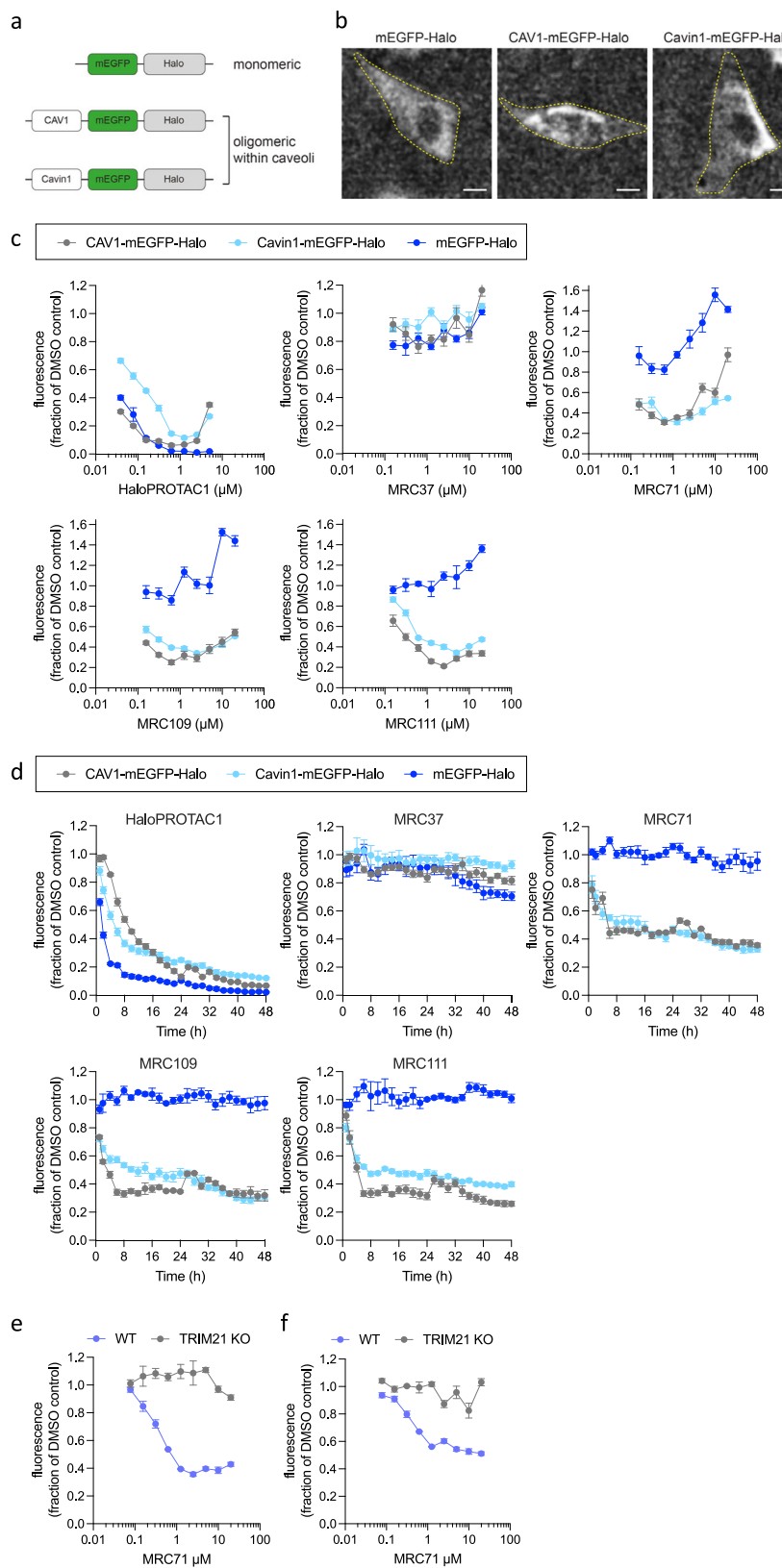

## TRIMTACs inhibit seeded tau aggregation

The aggregation of monomeric tau protein into elongated repetitive fibrils defines many common neurodegenerative diseases[45]. Small fragments of filaments may be released and taken up by neighbouring cells to template tau aggregation in recipient cells, in a process of seeded, or prion-like, aggregation. We have previously demonstrated that seeded aggregation can be intercepted by antibodies that rely on TRIM21 as their effector mechanism[17]. To examine whether seeded tau aggregation could be disrupted by a TRIMTAC in place of an antibody, we prepared filaments of full-length tau for the 0N4R isoform bearing the aggregation-prone P301S mutation and with a FKBP(F36V) domain fused at the N-terminus. FKBP(F36V)-Tau assemblies were incubated

**Fig. 3 | TRIMTACs degrade oligomeric but not monomeric targets. a** Three different Halo-tagged targets for TRIMTACs; mEGFP-Halo is a monomeric protein, CAV1- and Cavin1-mEGFP-Halo form oligomers within caveoli membrane structures. **b** Representative images of RPE-1 stable cell lines expressing the difference protein constructs; mEGFP-Halo exhibits diffuse cytosolic localisation, CAV1- and Cavin1-mEGFP-Halo exhibit a bright signal at the cell membrane indicative of caveoli structures. Yellow dotted line shows cell outline. Scale bar = 10 μm. **c** HaloTRIMTACs MRC71, 109 & 111 selectively degrade oligomeric CAV1- and Cavin1-mEGFP-Halo over monomeric mEGFP-Halo and exhibit a characteristic hook effect concentration-dependence for degradation. **d** Time course of degradation after treatment with compounds at 1.25 μM. **e, f** Degradation of oligomeric proteins (**e**) Cavin-1- and (**f**) CAV1-mEGFP-Halo HaloTRIMTAC MRC71 is dependent on TRIM21. **c–f** Graphs show the integrated density of GFP fluorescence (in relative fluorescence units, RFU) normalised to total cell area (phase) from $1 \times 10^4$ cells and expressed as a fraction of the DMSO control. Data is expressed as mean and s.e.m. from $n = 4$ technical replicates. Representative examples from $n = 2$ independent experiments. Source data are provided as a Source Data file.

with the dTAG-TRIMTAC compound MRC414 for 1 h prior to addition to HEK293 reporter cells expressing P301S tau-venus in the presence of lipofectamine. We observed that MRC414 was able to reduce seeded aggregation in a dose-dependent manner by approximately 60% (Fig. 6a, b). We observed that TRIM21 -only ligand MRC37 could compete with the activity of MRC414 to partially reverse its inhibition (Fig. 6c). In contrast to MRC414, we observed that a PROTAC equivalent, dTAG-VHL, was unable to significantly affect levels of seeded aggregation.

The utility of small molecule degraders relies on the ability to recruit endogenous ligases in disease-relevant cells. Previous studies using TRIMTACs have relied on ectopic over-expression of TRIM21 in immortalised cell lines in order to achieve degradation[27–30]. We therefore sought to repeat our tau degradation experiments in a primary neural culture setting. As the mouse TRIM21 PRYSPRY domain bears structural differences to the human version that could impact TRIMTAC binding, we developed a human TRIM21 knock-in mouse via gene replacement. Crucially, TRIM21 is expressed from the natural promoter and is therefore present at endogenous levels. We bred this line with tau transgenic Tg2541 mice that express human 0N4R tau bearing the P301S mutation under the control of a Thy1 promoter[46]. Ex vivo neuronal cultures prepared from these animals displayed dose-dependent intracellular tau aggregates (stained with the anti-phospho-tau antibody pS422) one week after challenge with FKBP(F36V)-tau assemblies, consistent with seeded tau aggregation (Supplementary Fig. 6). We next pre-incubated FKBP(F36V)-tau assemblies in the presence of MRC414 at 0.3 μM prior to addition to neural cultures. We observed that MRC414 reduced levels of induced hyperphosphorylated tau puncta and the number of cell bodies bearing abundant tau pathology (Fig. 6d–f). These results demonstrate that TRIMTAC molecules can substitute for antibodies and prevent the seeded aggregation of tau protein in disease-relevant primary cells[20,44].

## Discussion

PROTACs offer a therapeutic modality with the potential to not simply inhibit disease-causing proteins but remove them entirely[47]. They make use of degradation machinery that exists inside every cell, meaning that the range of potential disease targets is very broad[48]. Moreover, because PROTACs act catalytically, with each PROTAC molecule capable of facilitating the degradation of many target molecules, they have significant advantages over small-molecule inhibitors. Unlike inhibitors, PROTACs don't need to be dosed stoichiometrically and, because they aren't competing with natural ligands, there isn't the same requirement for high target affinity[49]. PROTACs manifest their own challenges, however, such as the difficulty of artificially inducing ubiquitin-mediated degradation through a targeting intermediary[35]. The design of PROTACs that co-opt classical CRLs such as VHL and CRBN has required, and continues to consume, huge chemistry resources. This has driven a search in recent years for alternative ligases and pathways that could be co-opted[50]. TRIM21 is a unique ligase in that it naturally works via a targeting intermediate, where the heterobifunctional molecule that binds both substrate and ligase is an antibody. Previously, we have shown that this allows antibodies to be used as 'bioPROTACs' to carry out targeted protein depletion in the technology 'Trim-Away'[28]. Alternative approaches

involving the direct fusion of TRIM21 catalytic domains with targeting domains such as nanobodies are also highly effective[31,33,34]. However, bioPROTACs have challenging drug properties, and this has driven the search for small-molecule ligands that can functionally replace antibodies.

Here we describe a small molecule ligand, MRC37, for TRIM21 that closely mimics the 'HNHY' epitope found in its natural IgG binding partner. Structural and biophysical studies confirm that MRC37 engages the same binding pocket as HNHY and interacts with many of the same TRIM21 residues in the PRYSPRY domain[4]. This binding activity allows MRC37 to inhibit TRIM21s natural antiviral function by preventing antibody neutralisation of adenovirus 5 (Adv5). Adenovirus is a widely used gene therapy vector, but previous studies have shown that TRIM21 uses anti-adenovirus antibodies to severely reduce in vivo gene delivery efficiency[51]. Use of an inhibitor based on MRC37 may therefore have utility as a treatment to mitigate against the problem of pre-existing immunity when using common viral serotypes or repeat treatments using vectors based on rare serotypes. MRC37 is also capable of inhibiting Trim-Away, a targeted protein degradation technology where TRIM21 is recruited to specific proteins via electroporated antibodies[15].

We have further shown that MRC37 is ideally suited as a warhead for making heterobifunctional molecules, as it possesses multiple attachment points where linkers can be added without altering TRIM21 binding. We have taken advantage of this to make a number of heterobifunctional degrader molecules, called "TRIMTACs". These TRIMTACs possess many of the same biophysical characteristics as PROTAC molecules: they are capable of forming ternary complexes between TRIM21 and neosubstrates, both in vitro and in cells. Like CRL-based ligands, TRIMTACs exhibit a 'hook effect' whereby excess ligand reduces ternary complex formation. During preparation of this manuscript, several studies were published that identified small-molecule degraders called molecular glues that form a complex between TRIM21 and Nup98, causing degradation of the latter[37–41]. When these glues were converted into heterobifunctional molecules, they were capable of degrading biomolecular condensates such as a chimeric PML fusion with EGFP and BRD4[37,38,40]. There are several key differences between these other reported ligands and MRC37. First, MRC37 does not induce TRIM21-dependent cytotoxicity, indicating that it is not a molecular glue between TRIM21 and Nup98. This is likely because MRC37 sterically inhibits such an interaction, a feature that may be advantageous for developing future degraders without this cytotoxic liability. Second, MRC37-based PROTACs induce degradation in unmodified cells using endogenous TRIM21, unlike PRLX93936 and Acepromazine-based degraders that require ectopic over-expression. Moreover, acepromazine-based degraders require expression of a D355A mutation because acepromazine engages only weakly with wild-type TRIM21[37].

In previous work, we have shown that TRIM21s clustering-based activation mechanism can be exploited to drive degradation of specific protein pools[20]. Taking advantage of the fact that TRIM21 will only activate when clustered by proteins that contain multiple repeats (such as mutant huntingtin), are highly oligomeric (such as Cavin1), or are aggregated (such as tau filaments), it is possible to selectively degrade these forms while leaving non-repetitive (wild-type

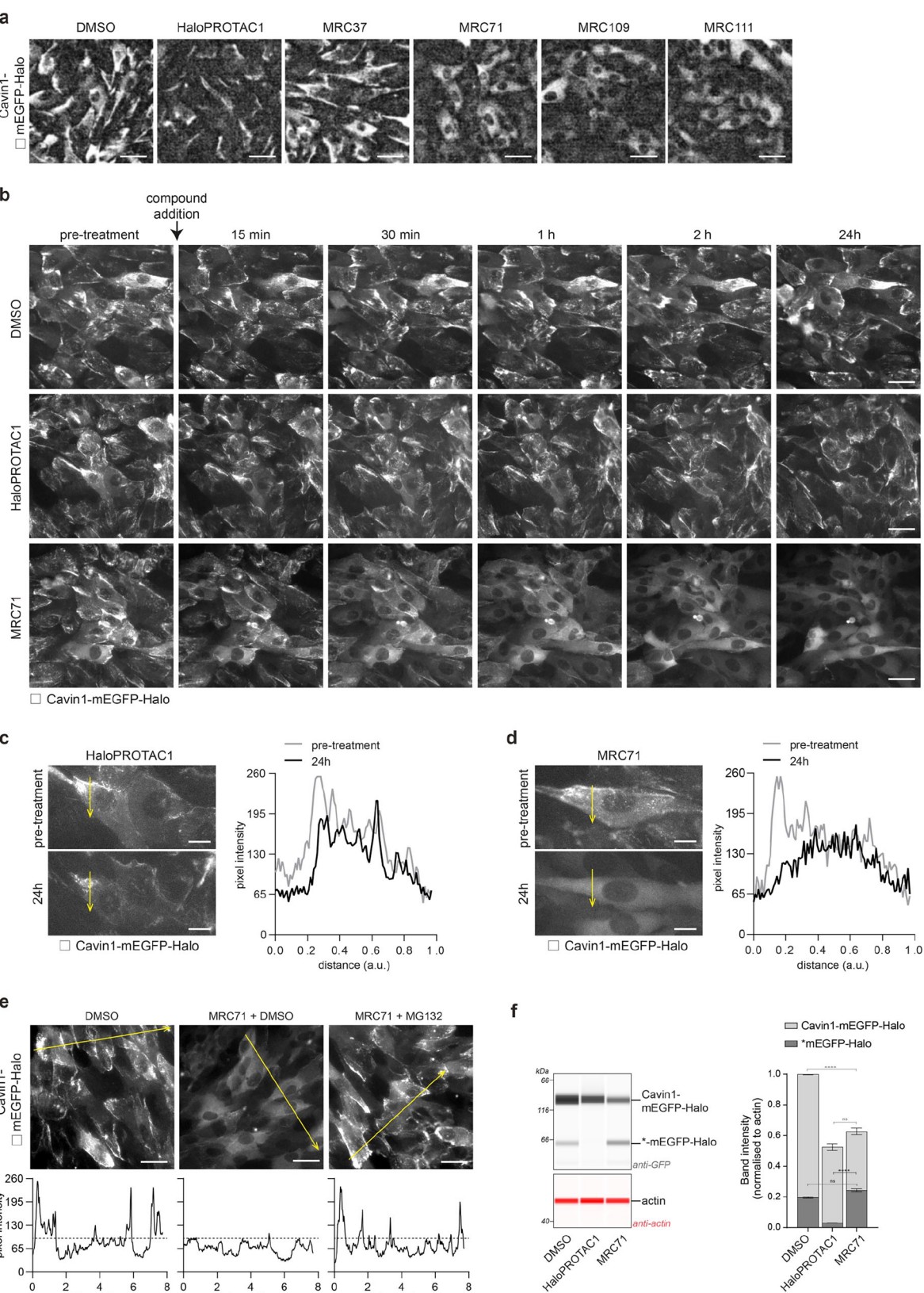

huntingtin), non-oligomeric (GFP) or non-aggregated (soluble tau) proteins untouched[20,28,30,33]. The small molecule TRIMTACs we describe here preserve the same requirement for TRIM21 clustering in order to drive degradation. MRC37-based TRIMTACs drive efficient degradation of a similar Cavin1 target as used previously in antibody-based Trim-away experiments[20]. Importantly, we show within the same

cell that membrane-associated oligomeric Cavin1 protein is degraded while monomeric cytoplasmic protein is not. In contrast, a VHL-based PROTAC degraded both membrane-associated and cytoplasmic protein pools. These experiments therefore highlight a key differentiator between PROTAC and TRIMTAC based degraders, with the latter possessing the unique ability to selectively degrade specific protein

**Fig. 4 | TRIMTACs are more efficient than PROTACs at degrading an oligomeric substrate. a** Cavin1-mEGFP-Halo exhibits a bright signal at the cell membranes and some diffuse cytoplasmic fluorescence in the control (DMSO) condition. 48 h treatment with HaloPROTAC1 (1.25 μM) reduces overall fluorescence, but residual fluorescence remains concentrated at the cell membranes. Treatment with Halo-TRIMTACs (1.25 μM) causes complete loss of bright fluorescence from cell membranes, but diffuse cytosolic fluorescence remains. Scale bar = 30 μm. **b** Treatment with HaloPROTAC1 causes a gradual loss of both cytosolic and membrane fluorescence over 24 h. Treatment with MRC71 causes rapid loss of membrane-associated fluorescence within 1 h. **c, d** Representative examples of individual cells from (**b**) pre-treatment and 24 h post-treatment with 2.5 μM HaloPROTAC1 or MRC71. Graphs show pixel intensities along the yellow line in images. Scale bar = 10 μm. **e** Cavin1-mEGFP-Halo cells imaged 2.5 h after treatment with DMSO or 2.5 μM MRC71 or 2.5 μM MRC71 + 25 μM MG132. Graphs show pixel intensities along the yellow line in images. Dashed line shows the peak maximum cytosolic pixel intensity; peaks above this line represent membrane-associated fluorescence. Rapid selective loss of membrane-associated fluorescence by MRC71 is prevented by MG132 treatment, suggesting proteasomal degradation. Scale bars = 30 μm. **f** Quantitative capillary-based ant-GFP immunoblotting of Cavin1-mEGFP-Halo cells following 24 h treatment with DMSO or 2.5 μM HaloPROTAC1 or MRC71. The two GFP species are indicated. Error bars depict mean ± s.e.m. from three technical replicates ($n = 3$). Significance based on two-way ANOVA with Tukey's multiple comparisons test. $P$-values for Cavin1-mEGFP-Halo are: DMSO vs MRC71 $p < 0.0001$, and for mEGFP-Halo are: HaloPROTAC1 vs MRC71 $p = 0.0006$. Source data are provided as a Source Data file.

pools depending on their assembly state within the same cell. Thus, it is possible to drive target specificity through the intrinsic catalytic mechanism of a ligase such as TRIM21 without needing to develop a state-specific warhead. Choice of TRIM21 ligand may still be important however. Degraders based on acepromazine and PRLX93936 were unable to degrade BRD4 unless it was part of a fusion protein like PML that formed a condensate. In contrast, a degrader based on HGC652 was capable of degrading unmodified BRD4[41]. BRD4 contains two BET domains, meaning that it is possible to recruit multiple TRIM21s even to monomeric BRD4. Moreover, as BRD4 associates with chromatin, there may be multiple adjacent BRD4 molecules that could mediate TRIM21 clustering upon its recruitment. The contrasting activities of the different reported degraders may be a result of their very different binding affinities, with the considerably higher affinity HGC652 ligand allowing degradation of a much less oligomeric target. We have previously noted a relationship between affinity and oligomeric degradation specificity when using a bioPROTAC based on the TRIM21 RING domain[33].

The ability to degrade state-specific protein pools allows for targeted protein degradation that removes disease-causing protein forms whilst leaving functional versions of the same protein untouched. Circumstances where this may be therapeutically beneficial include chronic inflammation driven by hyperactive signalling platforms or neurodegeneration caused by disease-causing protein aggregates. We have exemplified this here using model substrates Myd88, RIPK3 and tau. Myd88 is a key component of the myddosome, an oligomeric signalling platform that drives pro-inflammatory transcription[52]. TRIMTACs induced the selective degradation of Myd88 only when it had assembled into the myddosome, but not in unstimulated cells. RIPK3 is a cell-death regulator that binds and inhibits MLKL to suppress apoptosis[53], but when activated, assembles with RIPK1 into the necrosome, an oligomeric platform that drives necroptotic cell death[54]. Using a TRIMTAC, we were able to degrade RIPK3 selectively in cells undergoing necroptosis, partially rescuing cell death. Finally, TRIMTACs recapitulated the same inhibition of seeded tau aggregation as has been shown for antibodies. Protection was observed both in human reporter cell lines and in a neuronal model of seeded tau aggregation. Moreover, inhibition was observed under conditions where a related VHL-based PROTAC showed no activity. Taken together, these data show that our TRIMTAC degraders can discriminate between soluble and oligomeric forms of the same protein, suggesting that the clustering-based activation mechanism, which operates during TRIM21s natural function and in antibody-based Trim-Away, is maintained. Moreover, our results highlight that small molecule degraders like TRIMTACs, which utilise a non-CRL-based ligase, may have advantages over classic PROTACs. TRIMTACs therefore, hold promise as a potential strategy for selectively targeting cytosolic protein assemblies for modulation of cellular function, and to reduce aggregate load in neurodegenerative disease. The selectivity afforded by the TRIM21 mechanism suggests the strategy may find particular relevance in situations where removal of native protein is toxic due to an essential physiological role.

## Methods

### Mice
All animal work was licensed under the UK Animals (Scientific Procedures) Act 1986 and approved by the Medical Research Council Animal Welfare and Ethical Review Body. All work was carried out under establishment licence number X71139555 and PPL number PP109605 under MRC guidelines and regulations. P301S tau transgenic mice that had been extensively backcrossed to the C57BL/6 background were obtained from Prof Michel Goedert, Cambridge LMB (Tg2541). A hTRIM21 knockin model was obtained from Jackson labs. Animals were regularly monitored for clinical signs for the duration of all experiments. An equal number of male and female mice were used of P0 and P1 for ex vivo experiments and pooled for primary neuron cultures. Mice were kept in standard housing conditions with a 12 h light/dark cycle, an ambient room temperature of 22 °C and a humidity range of 40–60%.

### Plasmids
A full list of plasmids used in this study, including primary sequences of all constructs, can be found in (Supplementary Data 1). For lentivirus production, constructs were inserted into a modified version of pSMPP (Addgene #104970) where the SFFV promotor and puromycin resistance sequences were replaced with PGK1 promoter and Zeocin resistance sequences, respectively (pPMEZ). For protein purification, constructs were inserted into derivations of the pOP and pET (Novagen) series of vectors.

### General chemistry materials and methods
For full details of chemical materials and methods, please see the chemistry supporting information document. In brief, however, all reagents were purchased from commercial suppliers and used without further purification unless otherwise stated. For bespoke TRIM21 ligands and heterobifunctional compounds, synthesis was monitored by thin-layer chromatography (TLC) and NMR spectroscopy using a Bruker DPX-400 spectrometer. Compound purification was performed by semi-preparative reverse-phase HPLC on a Gilson HPLC system equipped with Gilson 306 pumps and a Phenomenex Synergi C18 (80 Å, 10 μm, 250 × 21.2 mm) column. Final compound purity was assessed by LCMS using an Agilent 6125B Single Quad LC-MS mass spectrometer with a 95% cut-off for purity.

### In vitro transcription of mRNA
pGEMHE plasmid constructs (Supplementary Data 1a) were linearised and 5′-capped mRNA was synthesised with T7 polymerase (NEB HiScribeT7 ARCA kit) according to the manufacturer's instructions. mRNA concentration was quantified using a Qubit 4 fluorometer (Thermo-Fisher) and RNA Broad Range assay kit (ThermoFisher; Q10211).

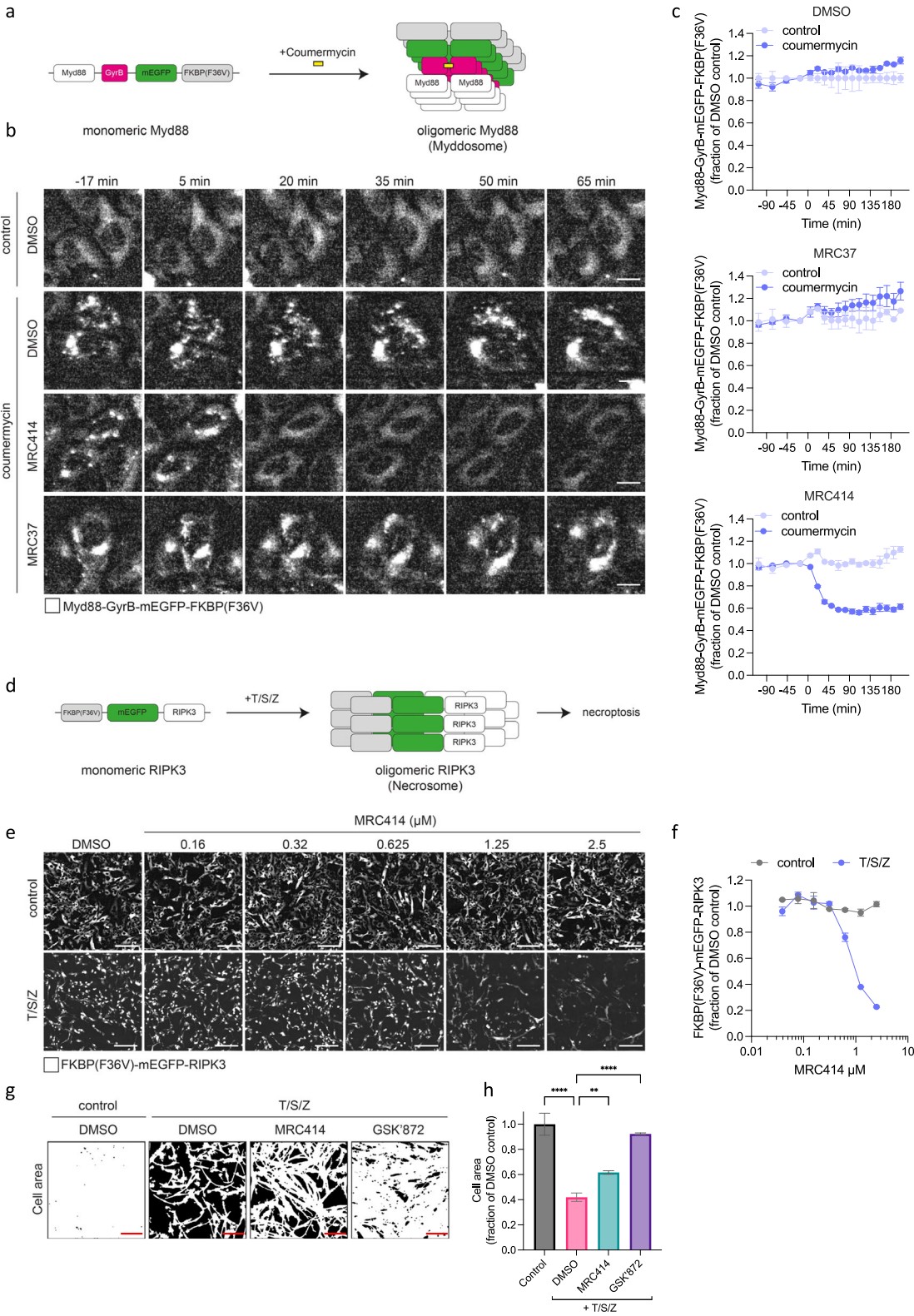

## Protein expression and purification

Proteins were recombinantly expressed in E. coli (C41) in 2XTY media (supplemented with 0.5% glucose, 2 mM MgSO4 and appropriate antibiotics) at 37 °C for 2–3 h (OD600 around 0.6-1) after which they were induced with 1 mM isopropylthio-β-galactoside (IPTG) and incubated at 18 °C overnight. Cells were pelleted with a Sorvall SLC-6000 compatible centrifuge at 4500 × g for 25 min and the pellet frozen until processed. The frozen pellet was resuspended in lysis buffer (50 mM Tris pH 8, 1 M NaCl, 10% v/v BugBuster (Novagen), 10 mM imidazole, 2 mM DTT and 1 × cOmplete protease inhibitors (Roche) and sonicated for 15 min total time (10 s on/20 s off) at 70% amplitude. Soluble fraction was recovered by centrifugation at 40 000 × g in a JLA25.50 rotor. The clarified lysate was applied to a gravity flow column prepared with 5–10 ml of NiNTA agarose (Qiagen) equilibrated with Buffer B

**Fig. 5 | TRIMTACs rapidly and selectively degrade active signalling complexes. a** Schematic of coumermycin-induced Myd88 oligomerization. **b**, **c** Live imaging and quantification of Myd88-GyrB-mEGFP-FKBP(F36V) fluorescence after treatment with DMSO or MRC37 or MRC414 at 5 μM. Compound addition at time 0. Scale bar = 10 μm. **d** Schematic of T/S/Z (TNF-α, SM-164 and zVAD-fmk) induced RIPK3 oligomerization. **e**, **f** Representative images and quantification of FKBP(F36V)-mEGFP-RIPK3 fluorescence 8 h post MRC414 addition at the indicated concentrations to either control or T/S/Z treated cells. Scale bar = 50 μm. **c**, **f** Graphs show the integrated density of GFP fluorescence (in relative fluorescence units, RFU) normalised to total cell area (phase) from $1 \times 10^4$ cells and expressed as a fraction of the DMSO control. Data is expressed as mean and s.e.m. from $n = 4$ technical replicates. Representative examples from $n = 2$ independent experiments. **g**, **h** Representative segmented cell area images and quantification at 48 h post-treatment. MRC414 was used at 2.5 μM and GSK'872 at 0.3 μM. Scale bar = 50 μm. Significance based on one-way Anova (*$P < 0.05$, ****$<0.0001$). Control vs DMSO $p < 0.0001$, DMSO vs MRC414 $p = 0.0053$, DMSO vs GSK'872 $p < 0.0001$. Graph shows the fraction of cell area as a proportion of the DMSO control in the absence of T/S/Z. Data is expressed as mean and s.e.m. from at least $n = 2$ technical replicates. Representative example from $n = 2$ independent experiments. Source data are provided as a Source Data file.

(300 mM NaCl, 50 mM Tris pH 8, 10 mM imidazole and 1 mM DTT). Bound fraction was washed in Buffer B (~1–20 bed volumes) and eluted with Buffer E (300 mM NaCl, 50 mM Tris pH 8, 400 mM imidazole and 1 mM DTT). Fractions of about 2 mL were collected for about 15–30 mL of eluate. Fractions containing the protein were pooled, filtered and separated by Size-Exclusion Chromatography (SEC) using HiLoad 26/600 Superdex 75/200 pg columns (Cytiva) in 150 mM NaCl, 50 mM Tris pH 8 and 1 mM DTT.

### Covalent capture of Trim21 PRYSPRY to magnetic beads
MagnaBind Carboxyl beads (Catalogue number: 21353) were diluted in water and washed. TRIM21his-PS was diluted to (0.1 mg/ml) in HEPES buffer (20 mM HEPES, pH 7.8, 150 mM NaCl and 1 mM TCEP) and stored on ice. The beads were activated as follows: 100 μL of both EDC (0.4 M) and NHS (0.1 M), which were mixed in a tube with 20 μL of 0.5 M MES buffer at pH 6 and the resulting mixture was incubated (20 °C, 1400 RPM) with the washed beads (100 μL per reaction) for 30 min. Next, the beads were washed to remove activated EDC and NHS and then incubated with the dilute solution of TRIM21his-PS for 15 min (20 °C, 1400 RPM). The unreacted products were quenched with 10 μL of 1 M Ethanolamine-HCl, pH 8.5 added directly to the mixture without washing. After a further 10 min, the solution was exchanged for fresh 50 mM Ethanolamine-HCl pH 8.5 and incubated for another 15 min. Finally, the beads were placed into HEPES buffer and were ready for use.

### Pulldown experiments with TRIM21hisPRYSPRY beads
TRIM21his-PS, no protein or control protein (C-terminal domain of SARS-CoV2 N protein) were covalently coupled to magnetic carboxyl-derived beads as described above. Binding experiments were carried out in 20 mM HEPES, pH 7.8, 150 mM NaCl and 1 mM TCEP. Beads were incubated with 2 μM of IgG (HB65) for 15 min, washed and the bound beads collected. Beads were incubated at 95 °C for 15 min to release bound proteins and analysed by SDS-PAGE and Coomassie staining. In the competition experiment, Protein AG (Pierce #21186) was included at 0.1, 1 or 10 μM during the binding incubation and otherwise processed as before.

### DEL screen selections
The selections were carried out according to the manufacturer's instructions: the DEL library was resuspended in screening buffer (1× PBS buffer supplemented with 0.05% v/v Tween 20 and 0.1 mg/mL sheared salmon sperm DNA) and incubated with fresh TRIM21his-PS beads for 1 h (20 °C, 1400 RPM). After 3 rounds of washing, the beads were boiled to release the bound molecules and incubated with a fresh batch of TRIM21his-PS beads. This process was repeated 3 times, and the final boiled sample was sent to WuxiAppTec for processing.

### Differential scanning fluorimetry
Thermal stabilisation assays were performed using the NanoTemper Prometheus NT48 instrument. All experiments were performed in 150 mM NaCl, 50 mM Tris pH 8, 1 mM DTT and 1% v/v DMSO unless otherwise noted. TRIM21his-PS was used at 5 μM and compounds at 100 μM. Samples were heated at 2 °C/min from 15 to 95 °C. Data were

analysed using the NanoTemper Prometheus Control Software and the first derivative of the melt curves was used to define the Tm for the apo and complexed samples.

### Crystallography and structural biology
TRIM21his-PS (15 mg/ml) was mixed 50:1 with a stock solution of compounds 36, 37 and 38 (100 mM in DMSO), left to equilibrate for 15–30 min before crystallisation plates were set up. Mosquito (TTP Labtech) instrument was used to set up sparse matrix screening, using a drop size of 100 nL protein with 200 nL of precipitant. Crystals were obtained by sitting vapour diffusion at 17 °C. Diffraction quality crystals for Compound 37 and Compound 38 were obtained in 30% PEG 10000, 0.1 M TRIS HCl, pH 8.5 and snap frozen in liquid nitrogen without cryoprotection. In the case of Compound 36, crystals were obtained in 20% PEG 4000, 5% iso-Propanol, 0.1 M Na Citrate, cryoprotected by supplementing the reservoir with 20% glycerol and snap-frozen in liquid nitrogen. For compound 209 TRIM21his-PS was diluted to 10 mg/ml and mixed 40:1 with a 50 mM stock of Compound 209 and crystallised in drops with 100 + 100 nL mixtures. Diffraction quality crystals were obtained in 20% (w/v) PEG-8000, 0.1 M CHES, pH 9.5 (final pH 9.6). Data were collected at the Diamond Light Source (Didcot, UK) on beamlines I-24 (for Compound 36, Compound 37 and Compound 209) and I-04 for Compound 38. HaloTagHis protein (13 mg/ml) was mixed 50:1 with a stock solution of Compound 71 (50 mM in DMSO), left to equilibrate for 30 min before crystallisation plates were set up. Mosquito (TTP Labtech) instrument was used to set up sparse matrix screening, using a drop size of 100 nL protein with 100 nL of precipitant. Crystals were obtained by sitting vapour diffusion at 17 °C. Diffraction quality crystals were obtained in 25% PEG 4000, 0.1 M Na MES, pH 6.5, 0.2 M Mg Chloride. Crystals were cryoprotected with reservoir solution supplemented with 20–30% glycerol and snap-frozen in liquid nitrogen. Data were collected at the Diamond Light Source (Didcot, UK) on beamline I-04. Data were processed with the CCP4i package as follows: Data were indexed, scaled and integrated using the Xia2dials (Winter et al., 2022) pipeline. Molecular replacement was performed with PHASER using the HaloTag model from 5UY1 or the TRIM21 PRYSPRY model from 2IWG. The model was iteratively refined using REFMAC5 and COOT for model building. AceDRG was used to generate restraints for ligands. PyMOL was used to generate all graphics involving structures.

### Modelling of ligands
Modelling of bound structures of ligands in the absence of experimental data was carried out using COOT. AceDRG was used to generate a PDB file of the compound of interest, and this was then manually docked in the electron density of a similar ligand. In the case of TRIM21, the data from the complex of TRIM21his-PS with MRC37 was used to dock compounds MRC71 and MRC73. To generate models of the ternary complex between TRIM21 and HaloTag, HADDOCK software was used to generate docked models of TRIM21his-PS and HaloTag. These were inspected for appropriate orientation, and then COOT and AceDRG were used to build the model of MRC71 in the ternary complex of TRIM21his-PS and HaloTag. All models were visualised using PyMOL.

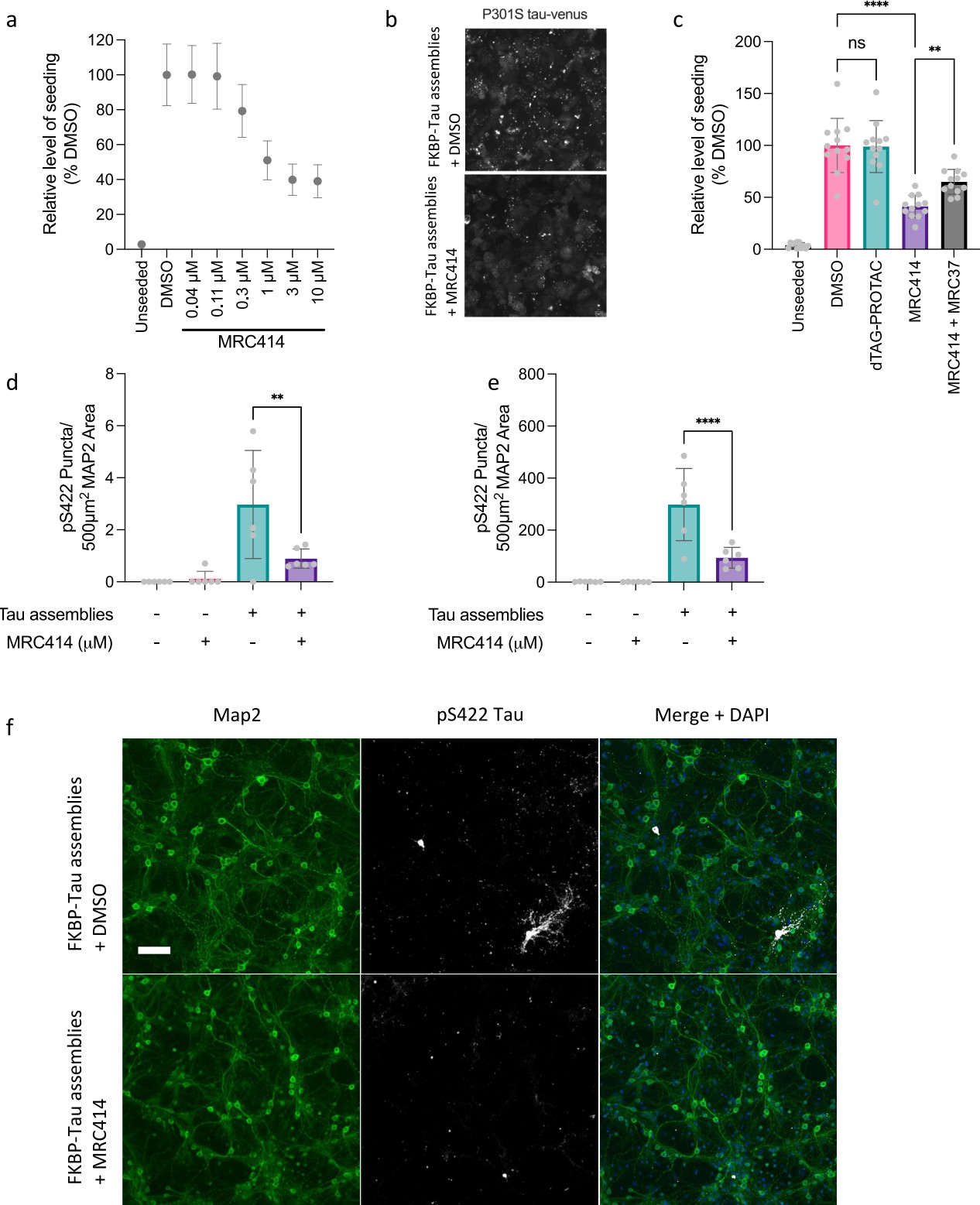

### Tryptophan quenching assay

Measured using nanoDSF Prometheus instrument at 15 °C (Compound 36) or a PHERAstar plate reader at 25 °C for the other two compounds (37 and 38). Assay was performed in 150 mM NaCl, 50 mM Tris pH 8, 1 mM DTT and 1% DMSO. Compound dilutions were prepared as required (usually 2-fold dilutions) starting from 100 μM. TRIM21his-PS was added from a 10X stock solution prepared in the same buffer for a final protein concentration of 8 μM for Prometheus experiments and 0.15 μM for PHERAstar experiments. Samples were prepared in a 384-well format (50 μL final volume in the well) or microtubes (40 μL reactions prepared) and left to equilibrate for 15 min, and then the fluorescence was measured. Data were normalised and analysed in PRISM.

**Fig. 6 | TRIMTACs can inhibit seeded tau aggregation. a** Levels of seeded aggregation in reporter HEK293 cells bearing P301S tau-venus. Cells were treated with FKBP(F36V)-tau assemblies in the presence of the indicated dose of dTAG-TRIMTAC prior to delivery of the assemblies to cells using lipofectamine. Seeded aggregation was recorded by Venus fluorescent spot detection 48 h after challenge. Error bars indicate mean ± SD from $n = 4$ biological repeats from $n = 3$ independent experiments. **b** Representative images of Venus fluorescence in cells treated as in panel (**a**) Scale bar = 10 μM. **c** Levels of seeded aggregation in HEK293 P301S tau-venus cells challenged with FKBP(F36V)-tau assemblies treated with DMSO alone, dTAG-VHL or dTAG-TRIMTAC with or without excess TRIM21 ligand MRC37. Error bars indicate mean ± SD from $n = 4$ biological repeats from $n = 3$ independent experiments. Significance based on one-way ANOVA with Sidak's adjustment for multiple comparisons (** $P < 0.01$; $P < 0.0001$). DMSO vs MRC414 $p < 0.0001$, MRC414 vs MRC414 + MRC37 $p = 0.0052$. **d** Number of cell bodies bearing pS422-positive tau assemblies in primary mouse neurons prepared from Tg2541 P301S tau transgenic mice crossed with human TRIM21 knockin animals. FKBP(F36V)-tau assemblies were incubated with the indicated concentration of dTAG-TRIMTAC prior to addition to neural cultures without any transfection reagent. Cultures were fixed 7 d post challenge with tau assemblies. pS422 signal was normalised to neuronal marker Map2. Significance based on one-way ANOVA with Sidak's adjustment for multiple comparisons (** $P < 0.01$), MRC414 vs DMSO $p = 0.0029$. **e** Levels of small tau puncta in neural cultures treated as in (**d**). **d**, **e** Error bars indicate mean ± SD from $n = 6$ biological repeats. Each point represents the mean value quantified from two images. Significance based on one-way ANOVA with Sidak's adjustment for multiple comparisons (**** $P < 0.0001$). MRC414 vs DMSO $p < 0.0001$. **f** Representative immunofluorescent images demonstrating pS422-positive tau structures formed in cultures from experiments as in (**d**). Scale bar represents 100 μm. Source data are provided as a Source Data file.

## Fluorescence polarisation experiment

Experiments were performed on a PHERAstar plate reader at 25 °C in 150 mM NaCl, 50 mM Tris pH 8, 1 mM DTT, 0.01% v/v tween20 and 1% DMSO in 384-well low binding plates. For competition experiments, TRIM21his-PS-Alexa488 (50 nM) was pre-mixed with human IgG Fc (5 μM) and left to equilibrate. 5 μL of this master mix was aliquoted into the 384-well plates. Compound dilutions were prepared in low binding 96 well plates, usually as 2-fold-dilutions starting from 10–100 μM. Next, 45 μL of the compound solutions was transferred to each well in the 384 well plate. The plate was centrifuged at 700 g for 1 min and left to equilibrate for 30 min in the dark. Finally, polarisation was read on the PHERAstar plate reader. Data were analysed in PRISM.

## Analytical ultracentrifugation

Velocity experiments were carried out using 12-mm two-sector cells, placed in an An-50 Ti rotor and centrifuged at 50,000 rpm at 20 °C using an Optima XL-I analytical ultracentrifuge (Beckman), and the samples were monitored using absorbance (A230 nm). For the ternary complex formation experiment, 5 μM of TRIM21his-PS (22,490 Da, calculated frictional ratio of 1.142) and 5 μM HaloTagHis (34,765 Da, calculated frictional ratio of 1.143) were present either alone, mixed in the presence of DMSO or mixed in the presence of 5–10 μM of compound MRC71. The data were analyzed in SEDFIT. The equilibrium experiment was performed with 5 μM of TRIM21his-PS and 5 μM HaloTagHis with either DMSO or a titration of MRC71 (2.5, 5, 10 or 20 μM) in 50 mM Tris HCl, pH 8.0, 150 mM NaCl, 1 mM DTT and 0.08 % (v/v) DMSO, were subjected to AUC sedimentation equilibrium at 20 ˚C using both absorbance at 280 nm and interference optics. The data were analysed in SEDPHAT using a single exponential function to obtain an average mass.

## Cell culture

Cell lines used and generated in this manuscript are detailed in Supplementary Data 1. Cells were maintained in DMEM (for HEK293T), DMEM/F-12 (for hTERT RPE-1) and McCoy's 5 A (for U2OS) supplemented with 10% (v/v) foetal calf serum (FCS), 100 U/mL penicillin, and 100 μg/mL streptomycin at 37 °C in a humidified atmosphere containing 5% $CO_2$ and regularly checked to be myoplasma-free. Live imaging was performed using the IncuCyte S3 live cell analysis system (Sartorius) and the Etaluma Lumascope LS720 widefield microscope equipped with a 40x apochromat 0.95NA air objective, both housed within a 37 °C, 5% $CO_2$ humidified incubator. For live imaging, cell culture medium was replaced with Fluorobrite (Gibco; A1896701) supplemented with 10% calf serum and GlutaMAX (Gibco; 35050061).

## Electroporation

Electroporation was performed using the Neon® Transfection System (Thermo Fisher). Cells were washed with PBS and resuspended in Buffer R at a concentration of between 1 - 8 x 10⁷ cells ml⁻¹. For each electroporation reaction, 1 - 8 x 10⁵ cells in a 10.5 μl volume were mixed with 2 μl of antibody (typically 0.5 mg/ml) or mRNA (typically 0.5 μM) or protein to be delivered. The mixture was taken up into a 10 μl Neon® Pipette Tip, electroporated at 1400 V, 20 ms, 2 pulses and transferred to media without antibiotics.

## Lentivirus production

Lentivirus particles were collected from HEK293T cell supernatant 3 days after co-transfection (FuGENE 6, Promega) of lentiviral plasmid constructs (Suppplementary Data 1) with HIV-1 GagPol expresser pcRV1 (a gift from Dr. Stuart Neil) and pMD2G, a gift from Didier Trono (Addgene plasmid #12259). Supernatant was filtered at 0.45 μm before storage at −80 °C.

## Cell Lines

Cell lines used and generated in this manuscript are detailed in (Supplementary Data 1). Cells were used expressing soluble human 0N4R P301S tau with a C-terminal Venus fluorophore[29]. RPE-1 WT and TRIM21 KO cells were also used[20]. Stable cell lines were generated by transducing cells with lentiviral particles at a multiplicity of ~0.1 transducing units per cell and selecting the GFP- and/or mCherry-positive populations by flow cytometry.

## Quantification of fluorescence in live cells

To quantify GFP fluorescence in live cells, images were acquired and analysed using the IncuCyte live cell analysis system (Sartorius). Within the IncuCyte software, the integrated density (the product of the area and mean intensity) for GFP fluorescence was normalised to total cell area (phase) for each image. Values were normalised to internal controls within each experiment. For the live cell ternary complex assay, ternary complex formation was quantified at 48 h post-compound addition by normalising the red/green overlap area (mCherry-TRIM21 positive nuclei) to total green area (total H2B-mEGFP nuclei).

## Adv5 neutralisation assay

1.0 x 10⁴ HEK293T cells were plated in 96-well format the day before infections. Cultures were pre-treated with TRIMTAC compounds at the indicated concentrations for 2 h prior to infections. Adenovirus serotype 5 2.6-del CMV-eGFP (Viraquest) was diluted to 1.1 × 10⁹ T.U./mL in PBS, and 16 uL was incubated 1:1 (v:v) with the indicated concentrations of anti-hexon recombinant humanised IgG1 9C12 or PBS. After 1 h incubation at room temperature, complexes were diluted with 250 μL Fluorobrite media. HEK293T cells were inoculated with 5 uL of viral complexes per well, and mixed gently. Infection was measured after 24 h by EGFP fluorescence using an Incucyte imager (Sartorius). EGFP-positive cells were counted and normalised to cell confluency and baseline infection.

## Myddosome assay

Expression construct comprises full-length Myd88 c-terminally fused to the GyrB domain of *Eschericia coli* DNA gyrase. The GyrB domain

dimerises upon binding to coumermycin, a bivalent antibiotic. Coumermycin-induced GyrB dimerisation induces Myd88 oligomerization and active Myddosome assembly. The mEGFP tag is included for visualisation. The FKBP(F36V) domain allows recruitment of TRIM21 via the MRC414 compound. U2OS cells expressing Myd88-GyrB-mEGFP-FKBP(F36V) were treated for 5 h with coumermycin (100 nM) to induce Myddosome assembly, or mock-treated (control). 5 h post-coumermycin/mock treatment, cells were treated with compounds, and GFP fluorescence was imaged and quantified using the IncuCyte system.

## Necrosome assay
Expression construct comprises full-length RIPK3 N-terminally fused to mEGFP for visualisation and FKBP(F36V) for TRIM21 recruitment via MRC414 compound. RPE-1 cells stably expressing FKBP(F36V)-mEGFP-RIPK3 were treated with T/S/Z (TNFα/Smac-mimetic/ZVAD-FMK) to induce RIPK3 oligomerization, or mock-treated (control), in the presence of a titration of compounds for 8 h, and GFP fluorescence was imaged and quantified using the IncuCyte system. TNFa was used at 20 ng/ml. Smac-mimetic (AZD5582) was used at 100 nM. ZVAD-FMK was used at 25 μM.

## Cell death assay
RPE-1 cells were plated in a 96-well plate (Corning 3595) at 20000 cells per well in DMEM/F12 (Gibco 10565018) media supplemented with GlutaMAX, FBS and Penicillin/Streptomycin. Cells were grown at 37 °C overnight. Next day, cells were treated with increasing concentrations of the compound in the media up to 25 μM. The dilution media was supplemented with DMSO to keep the concentration the same in all dilutions. Overnight media was removed and media with compound/DMSO was added to cells, in triplicate. Cells were imaged using Sartorius IncuCyte for 48 h, and cell confluency analyzed using IncuCyte software. Images were acquired every 1 h for the first 6 h and every 2 h up to 48 h. After 48 h, the cells were lysed using the detergent buffer from Luminescent ATP Detection Assay Kit (ab113849), the lysed cells were transferred into a white 96-well plate (Greiner 655075), and ATP levels was measured following manufacturer's protocol. The luminescence was read on the GloMax Discover plate reader (Promega). All results were normalised to the DMSO control, and Incucyte results were additionally normalised to time zero. Normalised results were plotted in Prism.

## Capillary-based immunoblotting
RIPA buffer protein extracts were diluted 1:2 in 0.1x sample buffer (biotechne; 042-195) and run on the Jess Simple Western system using a 12-230 kDa separation module (bio-techne) according to manufacturer's instructions. Antibodies and dilutions used for capillary-based immunoblotting (Jess) are detailed in (Supplementary Data 1). Protein peak areas were quantified using Compass software (biotechne) and normalised to internal protein loading controls within each capillary.

## Statistical analysis
Average (mean), standard deviation (s.d.), standard error of the mean (s.e.m.) and statistical significance based on Student's $t$-test (two-tailed) and one- or two-way ANOVA were calculated in Microsoft Excel or Graphpad Prism. Significance is sometimes represented with labels ns (not significant, $P > 0.05$), * ($P \leq 0.05$), ** ($P \leq 0.01$), *** ($P \leq 0.001$), **** ($P \leq 0.0001$).

## Preparation of and seeding with Sarkosyl-Insoluble FKBP-P301S Tau assemblies
HEK293 cells (15 cm dish) were transiently transfected using Lipofectamine™ 3000 with a construct expressing FKBP-P301S tau and sarkosyl-insoluble tau aggregates. At 72 h post-transfection, sarkosyl-insoluble FKBP-P301S tau assemblies were extracted using by lysed in ice-cold H-buffer (10 mM Tris, pH 7.4, 1 mM EGTA, 0.8 M NaCl, 10% w/v sucrose, supplemented with protease and phosphatase inhibitors), followed by five freeze–thaw cycles between −80 °C and room temperature. Lysates were clarified by centrifugation, then supplemented to a final sarkosyl concentration of 1%. Samples were then incubated at 37 °C for 1 h, before centrifuged at $100,000 \times g$ for 1 h at 4 °C. The resulting pellets were resuspended in TBS for use in experiments. For drug-treatment assays, tau assemblies were pre-incubated with test compounds for 1 h prior to transfection into cells expressing P301S tau-venus. Seeded aggregation was quantified 48 h post-transfection[29]. In competition experiments involving the T21 ligand, the target cells were pre-treated with the ligand for 24 h before addition of the drug–tau complex.

## Primary mouse neuron cultures
Brains were removed from the heads of 3 male and 3 female P0 and P1 C57BL/6 Tg2541/hTRIM21 mice, and pooled hippocampi and cortices were dissected in ice cold Hibernate-A (Gibco, A1247501), and the meninges removed. Tissues were pooled in a 15 mL conical tube and washed twice with room temperature Hibernate-A before being incubated with a final concentration of 0.25% trypsin (Gibco, 15090-046), at 37 °C for 20 min. During this period, a cotton-plugged glass Pasteur pipette (Merck Life Science, S6143) was fire-polished. Following trypsinisation, 500 μL 1% (w/v) DNAse I (Sigma-Aldrich, DN25) was added to the tissue and incubated at room temperature for 5 min. The tissue was washed twice with 37 °C Hibernate-A, followed by two washes with 37 °C neuron plating medium (PM) containing Neurobasal Plus (Gibco, A3582901), 1 mM GlutaMAX (Gibco, 15050061), 1% penicillin/streptomycin (Invitrogen, 15140122), 10 % horse serum (Invitrogen, 26050070), and 1x B-27 Plus supplement (Gibco, A352801). After washing, 2.5 ml PM was added to the tissue, and the tissue was triturated using the glass pipette in a 60 mm dish. A further 8 mL of PM was added to the dish, and the cell suspension was passed through a 70 μm cell strainer. Live cells were counted via trypan blue staining using the Countess II automated cell counter (Invitrogen). For tau seeding experiments, 45,000 cells/well were seeded into black 96-well plates (Greiner Bio-One, 655090), coated with poly-L-lysine (RnD Systems, 3438-100-01). After 4 h, all media was removed and replaced with maintenance media (MM) (PM without serum). All primary cultures were maintained in a humidified tissue culture incubator at 37 °C with 5% CO2.

## Seeded tau aggregation in primary mouse neurons
Seeded tau aggregation was induced in primary mouse neurons via the addition of $2.5 \times 10^{-4}$ μl FKBP-Tau P301S fibrils in maintenance media for seven days, followed by quantification via IF. Media was removed from neurons, followed by two washes with ice-cold PBS, after which neurons were fixed and permeabilised via the addition of 100 μL/well ice-cold methanol for 3 min on ice. Following this incubation, 100 μL PBS was added to dilute the methanol, and then 100 μL of this mixture was removed. This dilution step was repeated a total of three times, after which all liquid was removed from the well. Fixed neurons were then washed with PBS and blocked with 2% BSA in PBS (Neuron IF block) for 30 min at room temperature. Primary antibody solutions were made up in neuron IF block at the following concentrations: Rabbit anti-phospho-tau serine 422 (pS422) (Abcam, ab79415) 1:1000; Chicken anti-MAP2 12 (Abcam, ab5392) 1:5000. Neurons were incubated with primary antibody overnight at 4 °C. The following day, primary antibody was aspirated, and cells washed with PBS. Secondary antibody solutions were made up in neuron IF block at the following concentrations: goat anti-rabbit Alexa Fluor 647 (Invitrogen, A21245) 1:500; goat anti-chicken Alexa Fluor 488 (Invitrogen, A11039) 1:500. Neurons were incubated with secondary antibody for 1 h at room temperature in

the dark, after which the antibody was removed and neurons were washed with PBS. Finally, 2 µg/ml Hoechst 33342 diluted in PBS was added to neurons for 10 min at room temperature, followed by a final wash step in PBS. Cells were left in PBS for fluorescence imaging using an Eclipse Ti2 Microscope. NIS-Elements software (Nikon) was used to quantify pS422 puncta (puncta detection), pS422 cell bodies (thresholding) and MAP2 coverage in each image.

## Reporting summary

Further information on research design is available in the Nature Portfolio Reporting Summary linked to this article.

## Data availability

Structure coordinates have been deposited in the Protein Data Bank (PDB) under the following accession codes: [9Q9O], [9Q9P], [9Q9Q], [9Q9R], and [9R4O]. Source data are provided with this paper.

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

## Acknowledgements

This work was supported by a Welcome Trust Investigator Award (223054/Z/21/Z) and MRC (UK; U105181010) funding to LCJ and a Sir Henry Dale Fellowship to W.A.M., jointly funded by the Wellcome Trust and the Royal Society (grant 206248/Z/17/Z) and by the Lister Institute for Preventative Medicine. Further support was provided by the UK Dementia Research Institute, which receives its funding from DRI, funded by the UK Medical Research Council, Alzheimer's Society, and Alzheimer's Research UK. DJF and ACD acknowledge EPSRC grant EP/V008404/1 for support. Expertise in chemical synthesis was also provided by Syngene International.

## Author contributions

J.L., D.C., D.J.F., W.A.M. and L.C.J. conceived the study. L.C.J. wrote the manuscript. J.L., D.C., A.M., J.B., T.R., S.H.M., A.C.D., J.E.L. and M.S. performed the experiments.

## Competing interests

The authors declare the following competing interests: JL, DC, DJF and LCJ are inventors on a related patent application. WIPO (PCT) application filed by UKRI, inventors LCJ, DJF, DC and JL, WO2025021697A1, published 2025-01-30, PCT stage, covers MRC36-38. LCJ and WAM are co-founders of TRIMTECH Therapeutics, and DC and JB are TRIMTECH Therapeutics employees. TRIMTECH Therapeutics is a TRIMTAC protein degradation company. The remaining authors declare no competing interests.
