## [Transparent Peer Review file · Nature Communications]

State-selective small molecule degraders that preferentially remove aggregates and oligomers

Corresponding Author: Dr Leo James

Version 0:

Reviewer comments:

Reviewer #1

(Remarks to the Author)

This manuscript uses DNA-encoded library screening to identify novel ligands targeting the PRY-SPRY domain of TRIM21. While other TRIM21 PRY-SPRY ligands have been identified by similar means, those presented in this manuscript represent a unique structural class and demonstrate good affinity for their target. The authors then take a rigorous and systematic approach to PROTAC design, first obtaining structural data to demonstrate binding orientation and then extending linkers. The process of confirming binary complex binding, then demonstrating ternary complex binding in vitro before moving on to cells is rigorous.

Showing that a HaloTRIMTAC is more efficient at degrading a multimeric substrate relative to a HaloPROTAC targeting a widely used E3 ligase in the field (VHL) is a highlight. While previous work has shown that TRIMTacs spare monomeric substrates (Lu et al. Cell 2024), showing that TRIM21 is better than a widely used E3 like VHL further differentiates TRIM21 from other E3s for targeted protein degradation.

Degrading a therapeutically relevant oligomeric signaling complex composed of RIPK3 is another highlight and establishes proof-of-concept for other potential applications of TRIMTAC technology outside of the more established neurodegeneration space.

Finally, the authors demonstrate that they can inhibit seeded Tau aggregation in both an engineered HEK293T model and a mouse neuronal model. The more physiologically relevant neuronal model with endogenous TRIM21 is appreciated. Overall, this work is strong and will be of significant interest to those in the field of targeted protein degradation. This work expands the theoretical substrate scope for TRIM21 small molecule targeting, which remains a pressing question in the field. Both the rigor and breadth of the work, spanning structural biology, biochemistry, and cell biology, make this manuscript likely to be well suited for publication in Nature Communications.

1. Multiple other series of TRIM21 ligands have recently been shown to have molecular glue activity leading to degradation of nucleoporin proteins. Do the monomeric ligands identified from the DEL screen (or, less likely, resulting TRIMTACs) have this activity?

2. The introduction states that "GFP is not degraded in the presence of anti-GFP antibodies" and cites the original TRIM-away paper: Clift et al. Cell 2017. However, Figures 1 and 2 of that paper contradict this statement, showing that delivery of anti-GFP antibodies do degrade free GFP rapidly and efficiently. Can the author rationalize or clarify this apparent discrepancy?

3. "Leaves monomers untouched" in the title may be going a bit far. State-selective small molecule degraders that preferentially remove aggregates and oligomers?

4. There is no conflict of interest disclosure provided.

5. Recently published papers in the field should be referenced, particularly Zhang et al, JCI, published July 7. This paper shows degradation of monomeric BRD4 by TRIM21, which is a different model than presented here. It would be helpful to have the authors cite and discuss this new work.

Reviewer #2

(Remarks to the Author)

In this manuscript, the authors report the discovery of small molecule inhibitors of TRIM21, which they utilize to synthesise halo-tagged analogues, HaloTRIMTAC degraders that can selectively degrade polymeric proteins while leaving their monomeric counterparts untouched due to TRIM21's cluster-based activation mechanism. The authors also examine the

efficiency of the HaloTRIMTAC compared to a HaloVHL PROTAC and find the HaloTRIMTAC to be more efficient at degrading oligomeric species. The manuscript is well written, describing findings well-supported by robust data on a subject we believe will be of interest to the broad Nature Commun. readership. Hence, we recommend publication, after the comments outlined below are fully addressed.

Major Issues

1. The use of TRIMTAC to describe the compounds synthesized in the manuscript is misleading. We feel that designating the synthesized compounds as "TRIMTACs" is a misnomer, as there is no proximity-inducing small molecule described. The authors make extensive use of the HaloTag and dTAG systems to enable protein recruitment leading to degradation. Therefore, the authors should refer to the probes described in the manuscript as either HaloTRIMTACs or dTAGTRIMTACs, depending on the method of protein recruitment. Where appropriate the names should instead be changed to either HaloTRIMTAC or dTAGTRIMTAC
2. In the section "TRIMTACs are more efficient than PROTACs at degrading an oligomeric substrate" the authors only compare the HaloTRIMTAC against a VHL PROTAC. They should either make it clearer that HaloTRIMTAC is more efficient than HaloPROTAC (VHL) specifically, or evaluate at least another E3 ligase approach, e.g. HaloPROTAC (CRBN), in order to make a broader claim.
3. In figure 3c higher concentrations of MRC71 show a higher level of fluorescence from mEGFP-Halo. Please include a comment to explain this observation in the text or within the figure caption.

We also highlight several minor issues and suggestions:

1. Introduction – the abbreviation AAVs is used without being defined in the text
2. Results, section 1 – Please make it clear which of the binding methods referenced in the text (Supplementary figure 1b-e) were used to calculate the affinity between TRIM21 PRYSPRY and MRC36/37/38.
3. Figures 2a & 2b – Please increase the font size within the structures for clarity
4. Figure 3f – The labels for WT and TRIM21KO are the wrong way around
5. General chemistry methods and materials section is missing, and should be included.
6. There is no information on how was compound purity assessed.

Reviewer #3

(Remarks to the Author)

The manuscript titled "State-selective small molecule degraders that remove aggregates and oligomers but leave protein monomers untouched" from Luptak, Clift et al. presents the latest data from the James lab on the therapeutic potential of the ubiquitin E3 ligase TRIM21. Specifically, the authors show that TRIM21 can be targeted to oligomeric substrates in vitro and in cellulo using "TRIMTACs", that bind both TRIM21 and the substrate, leading to its degradation. Originally, this has only been possible using antibodies - the physiological substrate of TRIM21 - against the target (the Trim-Away system), though two recent studies have reported the identification of small molecule TRIMTACs, that enable degradation of neo-substrates (DOI: 10.1016/j.cell.2024.10.015 and <https://doi.org/10.1038/s41467-025-61818-7>).

Whilst the work described in this manuscript represents a potentially exciting step towards developing new treatments for diseases linked to protein aggregation, such as Alzheimer's, there are several points in this manuscript that should be improved, including providing more general context with respect to ubiquitination and PROTACs/molecular glues, providing a more detailed quantification and analysis of microscopy data and tone down some of the conclusions.

SPECIFIC POINTS

Comments on Figures

- The authors should more clearly specify the statistical methods used in each subfigure in the legend and whether mean +/- SD or SEM is used in each case, as well as the "n" of repeats. E.g. none of these are specified in the legend for Figure 5.
- For microscopy image quantification, please specify the total number of cells and/or images quantified in each analysis.

Figure 1

- "All three compounds competed with IgG Fc, suggesting an overlapping binding site" should refer to Supplementary Figure 1e, not d.

- The authors find that only W381, but not W383 and D355, in TRIM21 contributes to interactions with MRC37 and MRC38 (Supplementary Figure 1f). Could they suggest which other residues support to this single hydrophobic interaction, which alone wouldn't dictate selective binding? This table could also be labelled more clearly and the method of affinity analysis specified in the legend.

- Please specify the fluorescence units in the graphs Fig. 1d, e, g, h and Supplementary Fig. 1d.

Figure 2

- Error bars are missing for MRC71 and MRC209 in Fig. 2c.
- The interpretation of the diagram in Fig. 2f is not immediately obvious. Where wishing to indicate overlap this should be explicitly shown in the schematic.
- The mCherry-TRIM21, H2B-mEGFP-Halo co-localisation assay is used to demonstrate induced interaction mediated by MRC71 and used as a proxy for ternary complex formation (Fig. 2f-i). This is a convoluted assay and not one generally used in the PROTAC field. It would be more convincing to demonstrate this interaction in cells by using a co-immunoprecipitation or similar.
- The graphs in Fig. 2g-k appear to show differing levels of quantification for different TRIM21 constructs. mCherry-TRIM21 RB has approx. twice the peak T21-nuclei overlap area/nuclei area score of WT mCherry-TRIM21 following MRC71 treatment (0.09 vs 0.04) – could the authors please comment on why this might be? Is it an intrinsic variability of the assay or a property of the RB domains that favours TRIM21 cytosolic localisation?

- Please specify the units (e.g. μm^2 ?) for the axes in Fig. 2g, h, i, k.

Figure 3

- Please specify the fluorescence units in the graphs Fig. 3c-f.

- No scale bar in microscopy image in Supplementary Fig. 3.

Figure 4

- The quantification method in Figures 4c, 4d, and 4e are insufficient. The selective quantification of pixel intensity along the lines shown by yellow arrows is open to bias and at odds with the 'total' quantification levels used in the rest of the manuscript. Please re-analyse to include total quantification of mEGFP-Halo signal before and after treatments.

- Please specify the fluorescence units in the graphs in Supplementary Fig. 4c.

Figure 5

- The title of this section and its experimental conclusions are misleading – the authors do indeed show that they can induce degradation of oligomeric substrates (Myd88 and RIPK3), but they do not show that these assemblies are active signalling complexes. To support this conclusion, they would need to demonstrate TLR signalling activity or robust necrosis markers (beyond the cell area analysis in Fig. 5g) when Myd88 or RIPK3 oligomerisation are induced, respectively. Otherwise, please amend the wording to remove mention of “active signalling”.

- The authors need to specify which cell line is used in Fig. 5g.

- No scale bar given for microscopy images in Fig 5g.

- Please specify the fluorescence units in the graphs Fig. 5c and f.

- Please specify the area units in graph Fig. 5h.

Figure 6

- In the version of the manuscript provided to reviewers, there appears to be an error in the micron symbol where it should specify “scale bar represents 100 μm ” in the text.

- No scale bar is provided in Fig. 6b.

- Please specify the area units in graph Fig. 6d, e.

- It appears that in the final lines of the Results section before the Discussion, “Figure 6C&E” should actually be “Figure 6d-f”.

Comments Text

- In the first sentence of the introduction the authors state that E3 ligases label target proteins with ubiquitin for degradation in the proteasome or by autophagy. This is not correct as ubiquitination can result in diverse cellular outcomes, not just degradation (e.g. see Dikic & Schulman, NRCMB, 2023). The authors should reword this. Generally, the paper would benefit from a slightly broader introduction to the ubiquitin system and TRIM proteins.

- The authors have written in the introduction that “TRIM21 is continuously expressed in all cells”. Please provide a reference to support this statement. The currently available literature suggests TRIM21 expression is primarily induced in inflammatory/infection states.

- Furthermore, the authors state that TRIM21 exists in an inactive state due to autoinhibitory interactions between the RING and B-box domain that are relieved by phosphorylation. How is this phosphorylation triggered by TRIMTACs?

- The authors provide quite a limited set of references, largely covering their own previous work (19 of 36 references). The introduction and discussion of the manuscript would be more generally accessible and useful if the authors provided more context about PROTACs in general, as well as the molecular mechanisms and therapeutic potential of different E3 ligase families, especially as they refer to differences between CRLs and TRIM21.

- Figure references in the text use a mix of styles (e.g. “Figure 5E&F” and “Figure 5g,h”). These need to be unified.

- In the Results section titled “Small molecules can mimic antibody binding to TRIM21 and inhibit its antiviral and degradative activity”, the authors progressively focus in on MRC37. Please briefly explain how this focus evolved.

- Importantly, the authors need to stick to one nomenclature throughout the figures and text when referring to the chemical compounds. Currently it is difficult to follow with mixed use of MRC numbers and synonyms. For example, just in Figure 6 and its associated text there is interchangeable use of MRC414, dTAG-TRIMTAC, dTAG-TRIM21, and dTAG-TT, which presumably all refer to the same chemical moiety, as well as mixed use of dTAG-PROTAC and dTAG-VHL (N.B. VHL itself is not introduced as being a component of a CRL complex in this manuscript, a point which is important for the message of the paper).

- MRC37 is not a “warhead”, that description refers to an electrophile that is capable of forming a covalent bond with nucleophilic amino acids.

- The authors should comment on the most recent TRIMTAC publication in their discussion (<https://doi.org/10.1038/s41467-025-61818-7>).

Reviewer #4

(Remarks to the Author)

Reviewer #5

(Remarks to the Author)

I co-reviewed this manuscript with one of the reviewers who provided the listed reports. This is part of the Nature Communications initiative to facilitate training in peer review and to provide appropriate recognition for Early Career

Researchers who co-review manuscripts.

Version 1:

Reviewer comments:

Reviewer #1

(Remarks to the Author)
Publish without further changes.

Reviewer #2

(Remarks to the Author)
The authors response to my comments is adequate - thank you.

Reviewer #3

(Remarks to the Author)
The authors have made extensive edits to the manuscript to clarify the text and add requested information to figures and legends. This has addressed all our concerns.
We now fully support publication of this work in Nature Communications.

Reviewer #4

(Remarks to the Author)
I co-reviewed this manuscript with one of the reviewers who provided the listed reports. This is part of the Nature Communications initiative to facilitate training in peer review and to provide appropriate recognition for Early Career Researchers who co-review manuscripts.

Reviewer #5

(Remarks to the Author)
I co-reviewed this manuscript with one of the reviewers who provided the listed reports. This is part of the Nature Communications initiative to facilitate training in peer review and to provide appropriate recognition for Early Career Researchers who co-review manuscripts.

Reviewer #1

Q1. Multiple other series of TRIM21 ligands have recently been shown to have molecular glue activity leading to degradation of nucleoporin proteins. Do the monomeric ligands identified from the DEL screen (or, less likely, resulting TRIMTACs) have this activity?

A1. This is an interesting and important question. To address this, we have tested our best warhead, MRC37, in comparison to the identified molecular glue PRLX-93936. MRC37 did not show any cytotoxicity after 48hrs treatment at 20 μ M, either on-target (ie TRIM21-dependent) or off-target. In contrast, we observed TRIM21-dependent cytotoxicity for PRLX-93936 beginning at 12hrs treatment for concentrations above 1 μ M. MRC37 therefore does not appear to have the same glue liability as previously described TRIM21 compounds, possibly because MRC37 sterically inhibits a glue interaction with Nup98. We thank the reviewer for this excellent suggestion.

Q2. The introduction states that "GFP is not degraded in the presence of anti-GFP antibodies" and cites the original TRIM-away paper: Clift et al. Cell 2017. However, Figures 1 and 2 of that paper contradict this statement, showing that delivery of anti-GFP antibodies do degrade free GFP rapidly and efficiently. Can the author(s) rationalize or clarify this apparent discrepancy?

A2. This is indeed confusing as written and we thank the reviewer for picking this up. GFP can be degraded by antibodies but only when using a polyclonal mix, as in the 2017 paper. However, when a monoclonal anti-GFP antibody is used monomeric GFP cannot be degraded. The reason why polyclonal anti-GFP can induce degradation but not a monoclonal is because the polyclonal reagent contains a mix of antibodies that bind different epitopes, thus allowing multiple antibodies to bind per GFP monomer. This in turn allows multiple TRIM21 molecules to be recruited, which is a requirement for TRIM21 activation. The monoclonal vs polyclonal anti-GFP antibody data is not in Clift et al. Cell 2017 but in Zeng et al. NSMB 2021, which is the reference we should have cited in the original version of the manuscript. We have now added this correct reference to the revised manuscript.

Q3. "Leaves monomers untouched" in the title may be going a bit far. State-selective small molecule degraders that preferentially remove aggregates and oligomers?

A3. We have amended the title and used the recommended suggestion. This has also helped us to shorten it, thank you.

Q4. There is no conflict of interest disclosure provided.

A4. This has now been added.

Q5. Recently published papers in the field should be referenced, particularly Zhang et al, JCI, published July 7. This paper shows degradation of monomeric BRD4 by TRIM21, which is a different model than presented here. It would be helpful to have the authors cite and discuss this new work.

A5. We have expanded our discussion to include recent papers and highlight differences to our TRIMTACs:

"During preparation of this manuscript, several studies were published that identified small molecule degraders called molecular glues that form a complex between TRIM21 and Nup98, causing degradation of the latter[1-5]. When these glues were converted into heterobifunctional molecules, they were capable of degrading biomolecular condensates such as a chimeric PML fusion with EGFP and BRD4[1, 3, 4]. There are several key differences between these other reported ligands and MRC37.

First, MRC37 does not induce TRIM21-dependent cytotoxicity, indicating that it is not a molecular glue between TRIM21 and Nup98. This is likely because MRC37 sterically inhibits such an interaction, a feature that may be advantageous for developing future degraders without this cytotoxic liability. Second, MRC37-based PROTACs induce degradation in unmodified cells using endogenous TRIM21, unlike PRLX93936 and Acepromazine based-degraders that require ectopic overexpression. Moreover, acepromazine based-degraders require expression of a D355A mutation because acepromazine engages only weakly with wild-type TRIM21[3].

Choice of TRIM21 ligand may still be important however. A degrader based on acepromazine was unable to degrade BRD4 unless it was part of a fusion protein like PML that formed a condensate[3]. In contrast, a degrader based on HGC652 was capable of degrading unmodified BRD4[5]. BRD4 contains two BET domains, meaning that it is possible to recruit multiple TRIM21s even to monomeric BRD4. Moreover, as BRD4 associates with chromatin there may be multiple adjacent BRD4 molecules that could mediate TRIM21 clustering upon its recruitment. The contrasting activities of the different reported degraders may be a result of their very different binding affinities, with the considerably higher affinity HGC652 ligand allowing degradation of a much less oligomeric target. We have previously noted a relationship between affinity and oligomeric degradation specificity when using a bioPROTAC based on the TRIM21 RING domain.”

Reviewer #2:

Q1. The use of TRIMTAC to describe the compounds synthesized in the manuscript is misleading. We feel that designating the synthesized compounds as “TRIMTACs” is a misnomer, as there is no proximity-inducing small molecule described. The authors make extensive use of the HaloTag and dTAG systems to enable protein recruitment leading to degradation. Therefore, the authors should refer to the probes described in the manuscript as either HaloTRIMTACs or dTAGTRIMTACs, depending on the method of protein recruitment. Where appropriate the names should instead be changed to either HaloTRIMTAC or dTAGTRIMTAC

A1. We feel it is important to retain the name ‘TRIMTAC’ as the over-arching term for molecules that are proteolysis targeting chimeras and utilize TRIM21. Under this description, HaloTRIMTACs and dTAGTRIMTACs are types of TRIMTAC that utilize a Halo- or dTAG- binding ligand respectively. This follows the convention established by the term ‘PROTAC’ and where degraders based on VHL or CRBN that utilize Halo- or dTAG- binding ligands are types of PROTAC (eg as described in PMID: 36270231). However, we agree it is important that the reviewer know what type of moiety is being used to provide the substrate binding in a given experiment and so we have now ensured that the specific type of TRIMTAC, eg HaloTRIMTAC, is identified for each dataset.

Q2. In the section “TRIMTACs are more efficient than PROTACs at degrading an oligomeric substrate” the authors only compare the HaloTRIMTAC against a VHL PROTAC. They should either make it clearer that HaloTRIMTAC is more efficient than HaloPROTAC (VHL) specifically, or evaluate at least another E3 ligase approach, e.g. HaloPROTAC (CRBN), in order to make a broader claim.

A2. We have changed the name of the section to: “A haloTRIMTAC is more efficient than a VHL-based haloPROTAC at degrading an oligomeric substrate”. We have also clarified within the text of this section that we are using only a VHL-based degrader.

Q3. In figure 3c higher concentrations of MRC71 show a higher level of fluorescence from mEGFP-Halo. Please include a comment to explain this observation in the text or within the figure caption.

A3. We have added the following into the results text: "At high compound concentrations, particularly of MRC71, there was an increase in fluorescent monomeric mEGFP-Halo that is likely the result of binding-induced protein stabilization."

We also highlight several minor issues and suggestions:

Q1. Introduction – the abbreviation AAVs is used without being defined in the text.

A1. This has been corrected.

Q2. Results, section 1 – Please make it clear which of the binding methods referenced in the text (Supplementary figure 1b-e) were used to calculate the affinity between TRIM21 PRYSPRY and MRC36/37/38.

A2. This has been corrected to the following: "Three hits (MRC36, MRC37 and MRC38) were synthesized off-DNA and tested for binding to TRIM21 PRYSPRY by thermal stability and Tryptophan quenching, with fluorescence polarization used to determine accurate affinity measurements (Supplementary Figure 1b-e)."

Q3. Figures 2a & 2b – Please increase the font size within the structures for clarity.

A3. This has been corrected.

Q4. Figure 3f – The labels for WT and TRIM21KO are the wrong way around.

A4. Thank you for spotting - this has been corrected.

Q5. General chemistry methods and materials section is missing, and should be included.

A5. We have added a short chemistry section to the methods section and provided a detailed description of compound synthesis as a supplementary file.

Q6. There is no information on how was compound purity assessed.

A6. This has now been added to the chemistry methods section. In brief, all compounds were assessed by LCMS using an Agilent 6125B Single Quad LC-MS mass spectrometer with a 95% cut-off for purity.

Reviewer #3

Q1. The authors should more clearly specify the statistical methods used in each subfigure in the legend and whether mean +/- SD or SEM is used in each case, as well as the "n" of repeats. E.g. none of these are specified in the legend for Figure 5.

A1. Further details have been added to the figure legends in the revised manuscript.

Q2. For microscopy image quantification, please specify the total number of cells and/or images quantified in each analysis.

A2. This has been added to the figure legends.

Q3. Figure 1 - "All three compounds competed with IgG Fc, suggesting an overlapping binding site" should refer to Supplementary Figure 1e, not d.

A3. This has been corrected.

Q4. The authors find that only W381, but not W383 and D355, in TRIM21 contributes to interactions with MRC37 and MRC38 (Supplementary Figure 1f). Could they suggest which other residues support to this single hydrophobic interaction, which alone wouldn't dictate selective binding? This table could also be labelled more clearly and the method of affinity analysis specified in the legend.

A4. The table and legend have been updated and we have added the following to the results section: "While W381A is critical for MRC37 and MRC38 binding, there are other residues in proximity to the ligands that may contribute to binding, including Y328, F450, M330, S447, L371, L370, F369 and H368."

Q5. Please specify the fluorescence units in the graphs Fig. 1d, e, g, h and Supplementary Fig. 1d.

A5. The graphs in Figure 1 plot fluorescence as a fraction of 1 so don't have units. To clarify this we have added the following to the legend for these figures:

"Graphs show the integrated density of GFP fluorescence (in relative fluorescence units, RFU) normalized to total cell area (phase) from 1×10^4 cells and expressed as a fraction of the DMSO control."

The data in Supplementary Figure 1d is a percentage of fluorescence in the absence of compound. To clarify this we have added the following to the legend:

"Data is expressed as a percentage of fluorescence (RFU) in the absence of compound."

Q6. Figure 2 - Error bars are missing for MRC71 and MRC209 in Fig. 2c.

A6. The errors for these compounds are too small to be visible in the plot.

Q7. The interpretation of the diagram in Fig. 2f is not immediately obvious. Where wishing to indicate overlap this should be explicitly shown in the schematic.

A7. The diagram has been amended to show nuclear co-localization of red and green signals.

Q8. The mCherry-TRIM21, H2B-mEGFP-Halo co-localisation assay is used to demonstrate induced interaction mediated by MRC71 and used as a proxy for ternary complex formation (Fig. 2f-i). This is a convoluted assay and not one generally used in the PROTAC field. It would be more convincing to demonstrate this interaction in cells by using a co-immunoprecipitation or similar.

A8. As TRIM21 is an antibody binding protein, we prefer to avoid the use of immunoprecipitation as it can lead to confounding results. For instance as described here (PMID: 40216791). Co-localization of ligase and substrate by fluorescence microscopy is an accepted method in the field and we feel the data we present are unambiguous. Clear nuclear co-localization is seen for the heterobifunctional compound MRC71 but not for the TRIM21 only ligand MRC37.

Q9. The graphs in Fig. 2g-k appear to show differing levels of quantification for different TRIM21 constructs. mCherry-TRIM21 Δ RB has approx. twice the peak T21-nuclei overlap area/nuclei area score of WT mCherry-TRIM21 following MRC71 treatment (0.09 vs 0.04) – could the authors please comment on why this might be? Is it an intrinsic variability of the assay or a property of the RB domains that favours TRIM21 cytosolic localisation?

A9. This may be due to differences in expression levels of the two different constructs or in retention of the Δ RB protein in the nucleus. The Δ RB protein is also catalytically inactive and this could alter slightly its turnover in the presence or absence of compounds, though we think this latter explanation less likely. For the experiment in 2k, we used cells stably expressing both mCherry-TRIM21 and H2B-mEGFP-Halo (rather than transiently as in the other experiments) to increase the potential co-localization signal. We have now noted this in the results section.

Q10. Please specify the units (e.g. μm^2 ?) for the axes in Fig. 2g, h, i, k.

A10. The numbers on the axes are a ratio and don't have units. They represent the fraction of mCherry nuclear fluorescence that is also mEGFP fluorescent. This has been clarified in the figure legend.

Q11. Figure 3 - Please specify the fluorescence units in the graphs Fig. 3c-f.

A11. The fluorescence is a fraction of 1 and has no units. We have added the following explanation in the legend: "Graphs show the integrated density of GFP fluorescence (in relative fluorescence units, RFU) normalized to total cell area (phase) from 1×10^4 cells and expressed as a fraction of the DMSO control."

Q12. No scale bar in microscopy image in Supplementary Fig. 3.

A12. This has now been added.

Q13. Figure 4- The quantification method in Figures 4c, 4d, and 4e are insufficient. The selective quantification of pixel intensity along the lines shown by yellow arrows is open to bias and at odds with the 'total' quantification levels used in the rest of the manuscript. Please re-analyse to include total quantification of mEGFP-Halo signal before and after treatments.

A13. While we understand the reviewers concern in showing the fluorescence intensity of a cross-section, total quantification of the fluorescent signal would be misleading because it would not discriminate between the two populations of fluorescent substrate, membrane-associated and diffuse, in the cells. In addition to quantifying the fluorescence across the cross-section, we provide the corresponding microscopy images of the whole cell so it is clear to readers that the cross-section accurately reflects the distribution of fluorescent protein. Nevertheless, we agree that whole cell quantification of the different substrate levels is helpful, which is why we have provided this data in Figure 4f.

Q14. Please specify the fluorescence units in the graphs in Supplementary Fig. 4c.

A14. The fluorescence is a fraction of 1 and has no units. We have added the following explanation in the legend: "Graphs show the integrated density of GFP fluorescence (in relative fluorescence units, RFU) normalized to total cell area (phase) from 1×10^4 cells and expressed as a fraction of the DMSO control."

Q15. Figure 5 - The title of this section and its experimental conclusions are misleading – the authors do indeed show that they can induce degradation of oligomeric substrates (Myd88 and RIPK3), but they do not show that these assemblies are active signalling complexes. To support this conclusion, they would need to demonstrate TLR signalling activity or robust necrosis markers (beyond the cell area analysis in Fig. 5g) when Myd88 or RIPK3 oligomerisation are induced, respectively. Otherwise, please amend the wording to remove mention of "active signalling".

A15. We have removed the term active signalling and reworded this section.

Q16. The authors need to specify which cell line is used in Fig. 5g. No scale bar given for microscopy images in Fig 5g. Please specify the fluorescence units in the graphs Fig. 5c and f. Please specify the area units in graph Fig. 5h.

A16. Details of the cell line used in Figure 5 are given in the methods section: “Necrosome assay: Expression construct comprises full length RIPK3 N-terminally fused to mEGFP for visualisation and FKBP(F36V) for TRIM21 recruitment via MRC414 compound. RPE-1 cells stably expressing FKBP(F36V)-mEGFP-RIPK3 were treated with T/S/Z (TNF α /Smac-mimetic/ZVAD-FMK) to induce RIPK3 oligomerization, or mock treated (control), in the presence of a titration of compounds for 8h and GFP fluorescence imaged and quantified using the IncuCyte system. TNF α was used at 20 ng/ml. Smac-mimetic (AZD5582) was used at 100 nM. ZVAD-FMK was used at 25 μ M.”

A scale bar has been added to 5g. The fluorescence in graphs 5c and 5f are shown as a fraction of 1 and have no units. We have added the following explanation in the legend: “Graphs show the integrated density of GFP fluorescence (in relative fluorescence units, RFU) normalized to total cell area (phase) from 1×10^4 cells and expressed as a fraction of the DMSO control.” The graph in 5h doesn't have area units but is a fraction of the DMSO control in the absence of T/S/Z. We have added an explanation in the legend.

Q17. Figure 6 - In the version of the manuscript provided to reviewers, there appears to be an error in the micron symbol where it should specify “scale bar represents 100 μ m” in the text. No scale bar is provided in Fig. 6b. Please specify the area units in graph Fig. 6d, e.

A17. This has been corrected and area units provided.

Q18. It appears that in the final lines of the Results section before the Discussion, “Figure 6C&E” should actually be “Figure 6d-f”.

A18. This has been corrected.

Comments Text

Q1. In the first sentence of the introduction the authors state that E3 ligases label target proteins with ubiquitin for degradation in the proteasome or by autophagy. This is not correct as ubiquitination can result in diverse cellular outcomes, not just degradation (e.g. see Dikic & Schulman, NRMCB, 2023). The authors should reword this. Generally, the paper would benefit from a slightly broader introduction to the ubiquitin system and TRIM proteins.

A1. We have reworded this as requested and expanded the first introductory paragraph to include an overview on TRIM proteins, with accompanying references. We have also added a paragraph to the discussion to expand on the use of CRLs by PROTACs.

Q2. The authors have written in the introduction that “TRIM21 is continuously expressed in all cells”. Please provide a reference to support this statement. The currently available literature suggests TRIM21 expression is primarily induced in inflammatory/infection states.

A2. We have amended this sentence to: “TRIM21 is broadly expressed in many cell types and tissues”. We also now include three references in support of this statement, including the Human Protein Atlas that indicates broad protein and RNA expression.

Q3. Furthermore, the authors state that TRIM21 exists in an inactive state due to autoinhibitory interactions between the RING and B-box domain that are relieved by phosphorylation. How is this phosphorylation triggered by TRIMTACs?

A3. We state that TRIM21 is regulated by autoinhibitory interactions between RING and B-box domain. We don't think autoinhibition is relieved by phosphorylation but by the clustering mechanism reported in Zeng et al. 2021 NSMB. Consistent with this mechanism, we only see TRIMTAC-induced degradation of oligomeric but not monomeric substrates.

Q4. The authors provide quite a limited set of references, largely covering their own previous work (19 of 36 references). The introduction and discussion of the manuscript would be more generally accessible and useful if the authors provided more context about PROTACs in general, as well as the molecular mechanisms and therapeutic potential of different E3 ligase families, especially as they refer to differences between CRLs and TRIM21.

A4. We have added new sentences to the introduction and a new paragraph to the discussion. We completely agree with the reviewer that comparing the potential of different E3 ligase families to be recruited by small molecules for induced degradation is a really important and interesting question, though we have limited space to address this in the present manuscript. It would make a great topic for a review.

Q5. Figure references in the text use a mix of styles (e.g. "Figure 5E&F" and "Figure 5g,h"). These need to be unified.

A5. This has been corrected.

Q6. In the Results section titled "Small molecules can mimic antibody binding to TRIM21 and inhibit its antiviral and degradative activity", the authors progressively focus in on MRC37. Please briefly explain how this focus evolved.

A6. We have added the following sentence: "We chose to focus on MRC37 as this had the highest binding affinity for TRIM21 PRYSPRY and because we identified three possible exit vectors using our crystal structure where additional moieties could be easily added."

Q7. Importantly, the authors need to stick to one nomenclature throughout the figures and text when referring to the chemical compounds. Currently it is difficult to follow with mixed use of MRC numbers and synonyms. For example, just in Figure 6 and its associated text there is interchangeable use of MRC414, dTAG-TRIMTAC, dTAG-TRIM21, and dTAG-TT, which presumably all refer to the same chemical moiety, as well as mixed use of dTAG-PROTAC and dTAG-VHL (N.B. VHL itself is not introduced as being a component of a CRL complex in this manuscript, a point which is important for the message of the paper).

A7. We have simplified the nomenclature to refer to either the compound number or a single synonym that explains the compound type, for instance: "the dTAG-TRIMTAC compound MRC414". We hope this will allow the reader to easily follow which compound is being used.

Q8. MRC37 is not a "warhead", that description refers to an electrophile that is capable of forming a covalent bond with nucleophilic amino acids.

A8. We have replaced instances of the word warhead with ligand.

Q9. *The authors should comment on the most recent TRIMTAC publication in their discussion (<https://doi.org/10.1038/s41467-025-61818-7>).*

A9. We have expanded our discussion in light of recent publications as follows:

“During preparation of this manuscript, several studies were published that identified small molecule degraders called molecular glues that form a complex between TRIM21 and Nup98, causing degradation of the latter[39, 50-53]. When these glues were converted into heterobifunctional molecules, they were capable of degrading biomolecular condensates such as a chimeric PML fusion with EGFP and BRD4[39, 50, 52]. There are several key differences between these other reported ligands and MRC37. First, MRC37 does not induce TRIM21-dependent cytotoxicity, indicating that it is not a molecular glue between TRIM21 and Nup98. This is likely because MRC37 sterically inhibits such an interaction, a feature that may be advantageous for developing future degraders without this cytotoxic liability. Second, MRC37-based PROTACs induce degradation in unmodified cells using endogenous TRIM21, unlike PRLX93936 and Acepromazine based-degraders that require ectopic overexpression. Moreover, acepromazine based-degraders require expression of a D355A mutation because acepromazine engages only weakly with wild-type TRIM21[39].

Choice of TRIM21 ligand may still be important however. Degradation based on acepromazine and PRLX93936 were unable to degrade BRD4 unless it was part of a fusion protein like PML that formed a condensate. In contrast, a degrader based on HGC652 was capable of degrading unmodified BRD4[53]. BRD4 contains two BET domains, meaning that it is possible to recruit multiple TRIM21s even to monomeric BRD4. Moreover, as BRD4 associates with chromatin there may be multiple adjacent BRD4 molecules that could mediate TRIM21 clustering upon its recruitment. The contrasting activities of the different reported degraders may be a result of their very different binding affinities, with the considerably higher affinity HGC652 ligand allowing degradation of a much less oligomeric target. We have previously noted a relationship between affinity and oligomeric degradation specificity when using a bioPROTAC based on the TRIM21 RING domain[35].”